# Loss of Myo19 increases metastasis by enhancing microenvironmental ROS gradient and chemotaxis

Xiaoyu Ren[1,4], Peng Shi [1,2,4✉], Jing Su[3], Tonghua Wei[1], Jiayi Li[1], Yiping Hu[1] & Congying Wu [1,2✉]

## Abstract

**Tumor metastasis involves cells migrating directionally in response to external chemical signals. Reactive oxygen species (ROS) in the form of $H_2O_2$ has been demonstrated as a chemoattractant for neutrophils but its spatial characteristics in tumor microenvironment and potential role in tumor cell dissemination remain unknown. Here we investigate the spatial ROS distribution in 3D tumor spheroids and identify a ROS concentration gradient in spheroid periphery, which projects into a $H_2O_2$ gradient in tumor microenvironment. We further reveal the role of $H_2O_2$ gradient to induce chemotaxis of tumor cells by activating Src and subsequently inhibiting RhoA. Finally, we observe that the absence of mitochondria cristae remodeling proteins including the mitochondria-localized actin motor Myosin 19 (Myo19) enhances ROS gradient and promotes tumor dissemination. Myo19 downregulation is seen in many tumors, and Myo19 expression is negatively associated with tumor metastasis in vivo. Together, our study reveals the chemoattractant role of tumor microenvironmental ROS and implies the potential impact of mitochondria cristae disorganization on tumor invasion and metastasis.**

**Keywords** Myo19; Mitochondria; Chemotaxis; Tumor Metastasis; ROS Gradient
**Subject Category** Cancer

## Introduction

Deregulated cellular metabolism is a hallmark of cancer (Hanahan and Weinberg, 2011). Reactive oxygen species (ROS) are seen to be increased in many tumors (Szatrowski and Nathan, 1991), and have been suggested to regulate tumor metastasis (Cheung and Vousden, 2022). One major source of cellular ROS is electron leak during oxidative phosphorylation (OXPHOS), which takes place at mitochondria cristae in mammalian cells (Murphy, 2009; Sullivan and Chandel, 2014). The absence of proteins that remodel cristae such as Mic60 and Opa1 impacts OXPHOS (Cogliati et al, 2013; Friedman et al, 2015) and enhances ROS

production (John et al, 2005; Varanita et al, 2015). Accumulating evidence suggests that loss of cristae integrity is related to tumor metastasis. Mic60 depletion enhances lung metastatic foci in vivo (Ghosh et al, 2022), while degradation of Opa1 by mitochondria protease OMA1 promotes colorectal cancer development (Wu et al, 2021). Previously we and others have reported that mitochondria-localized actin motor Myo19 regulates cristae architecture through mechanical instability (Shi et al, 2022) and its deficiency impairs OXPHOS while enhances ROS production (Majstrowicz et al, 2021). Myo19 downregulation is seen in invasive cancers such as diffuse-type gastric carcinoma (Ge et al, 2018), hinting at a potential role of Myo19 loss in tumor invasiveness.

Compared to random motility, tumor invasion is more efficient when cells move in a directed mode such as chemotaxis (Roussos et al, 2011). $H_2O_2$ is a relatively stable ROS species and diffuses more rapidly (Bienert et al, 2006), allowing it to be a promising candidate to convey signals through the extracellular space of tissues and cells. It has been reported that neutrophils can migrate toward $H_2O_2$ in vitro and in vivo (Klyubin et al, 1996; Niethammer et al, 2009). $H_2O_2$ produced by the dual oxidase in the wound in zebrafish larvae can mediate rapid recruitment of neutrophils to the injury sites through activation of Lyn, a member of the Src family kinases (Niethammer et al, 2009; Tauzin et al, 2014; Yoo et al, 2012; Yoo et al, 2011). Since tumor cells and infiltrating immune cells experience similar chemical microenvironment, it is intriguing to investigate whether tumor cell migration can also be regulated by $H_2O_2$.

In this work, we investigated the spatial ROS distribution in tumor spheroids and identified a ROS concentration gradient in spheroid periphery, which could project into $H_2O_2$ gradient in the microenvironment. Functionally, extracellular $H_2O_2$ gradient could induce chemotaxis in tumor cells by activating Src and subsequently inhibiting RhoA, which contributed to ROS gradient-induced spheroid invasion. Finally, we found that the absence of mitochondria cristae remodeling proteins such as Myo19 and Mic60 enhanced spheroid ROS gradient by increasing mitochondria ROS, which was associated with a higher tendency of metastatic dissemination in vivo. Together, our observations highlight the important role of microenvironmental redox regulation in cancer progression, and provide new insights into the pathophysiological functions of mitochondria cristae remodeling proteins.

[1]Institute of Systems Biomedicine, Peking University Health Science Center, Key Laboratory of Tumor Systems Biology, Beijing 100191, China. [2]International Cancer Institute, Peking University, Beijing 100191, China. [3]Department of Pathology, School of Basic Medical Sciences, Peking University Third Hospital, Peking University Health Science Center, Beijing 100191, China. [4]These authors contributed equally: Xiaoyu Ren, Peng Shi. ✉E-mail: peng@bjmu.edu.cn; congyingwu@hsc.pku.edu.cn

# Results and discussion

## Myo19 downregulation increased tumor invasion and metastasis

In the EMBL-EBI (EMBL's European Bioinformatics Institute) database, downregulation in Myo19 mRNA was seen in multiple tumors, among which breast carcinoma displayed the most significant reduction of Myo19 mRNA (Fig. EV1A). Lymphatic metastasis is commonly seen in breast carcinoma patients with distant metastasis (To et al, 2020). Using immunohistochemical (IHC) staining of Myo19 in breast carcinoma samples (Fig. 1A), we detected decreased Myo19 expression in tumors with lymphatic metastasis (Fig. 1B), while no significant difference was found between Myo19 and Ki-67 expression, a maker for tumor cell proliferation (Menon et al, 2019) (Fig. EV1B). We further divided the samples into three groups according to Myo19 expression (Myo19 Low, Moderate and High). We detected higher frequency of lymphatic metastasis in patients with low Myo19 expression (17 metastatic incidents in 41 patients) compared to moderate expression group (6 metastatic incidents in 23 patients) (Fig. 1C), while no significance was found in Ki-67 expression between these two groups (Fig. EV1C,D). To evaluate the potential role of Myo19 in tumor dissemination, we injected 4T1 mouse breast carcinoma cells into the mouse mammary fat pads, and assessed tumor cells infiltrating into adjacent normal tissues including skeletal muscles, dermis, and epidermis by hematoxylin-eosin staining (HE) staining (Fig. EV1E,D). Interestingly, Myo19 knockdown (KD) led to higher invasiveness (Fig. 1E,F), without changes in tumor weight or the maximal tumor diameter (Fig. EV1F,G). In addition, increased lung metastatic foci were also detected in Myo19 KD tumors (Fig. 1G,H).

Previously we have reported that loss of Myo19 disrupts mitochondria cristae and hampers OXPHOS (Shi et al, 2022). Aberrant cristae morphology was observed in Myo19 deficient MDA-MB-231 cells (Fig. EV1H). To further elucidate the role of cristae remodeling proteins in tumor invasion, we assayed the growth and infiltration in 4T1 tumors with knockdown of Mic60, a critical subunit of the SAM-MICOS cristae remodeling complex at the cristae junction, loss of which also disrupted cristae integrity (Fig. EV1H) (John et al, 2005). Similarly, Mic60 downregulation also enhanced local invasion and lung metastasis (Fig. 1E,H) without tumor weight or maximal diameter alterations (Fig. EV1F,G). These results revealed that the downregulation of cristae remodeling proteins including Myo19 and Mic60 could promote tumor invasion and metastasis.

## Loss of cristae remodeling proteins enhanced spheroid ROS gradient

The absence of cristae sculpturing proteins disrupts cristae integrity, which enhances electron leakage and promotes ROS production (John et al, 2005; Varanita et al, 2015). Consistent with previous findings (Majstrowicz et al, 2021), we found Myo19 deficient cells displayed higher mitochondria matrix and inter-membrane space ROS (Fig. EV2A–C). Loss of Myo19 elevated the superoxide anion and $H_2O_2$ level in mitochondria (Fig. EV2D–G), as well as in the whole cell (Fig. EV2H–J). To investigate the ROS distribution in 3D system, we generated tumor spheroids with MDA-MB-231 human breast cancer cells expressing the RoGFP probe, a ratiometric fluorescent probe that is sensitive to redox stimulation (Hanson et al, 2004). Consistent with previous studies (Schafer et al, 2009; Takahashi et al, 2018), elevated ROS level in the inner core was detected (Fig. 2A). Interestingly, we also detected enhanced ROS level in spheroid outer shell, while the middle layer exhibited lower ROS level (Fig. 2A). Such ROS distribution at spheroid periphery, where dynamic cell motion took place, established a ROS concentration gradient (abbreviated as ROS gradient thereafter). To quantify this gradient, we created a series of inward scaling masks from the outline of the spheroid using the ROI manager (ImageJ) (Fig. 2B). The average RoGFP ratio of each concentric region was plotted as a function of its outer radius (Fig. 2C). The slope of the line connecting the highest point in the outer layer and the lowest point in the middle layer was then calculated as the ROS gradient. With similar setting, we also observed the existence of peripheral ROS gradient in HeLa, MCF7 and B16-F10 tumor spheroids (Fig. 2D,E). Downregulation of Nrf2 (Fig. EV2K), the major antioxidant transcription factor in tumor cells (Vomund et al, 2017), could blunt the ROS gradient by elevating the ROS level in the middle layer to a larger extent than that in the outer layer (Fig. EV2L,M), possibly due to the fact that Nrf2 is less activated in the outer shell of tumor spheroids (Kipp et al, 2017).

Next, we sought to explore whether Myo19 deficiency could affect ROS gradient. In Myo19 KO spheroids, ROS level of spheroid outer layer increased to a larger extent than that in the middle layer (Fig. EV2O), and ROS gradient was significantly enhanced (Fig. 2F). These observations could be recapitulated with Mic60 knockdown, which was previously reported to increase mitochondria ROS (John et al, 2005) (Figs. 2G and EV2N,P). When we applied Antimycin A (AA) to increase mitochondria ROS by promoting electron leak (Park et al, 2007), we observed most prominently increased ROS level in the outer layer (Fig. EV2Q), resulting in a sharper peripheral ROS gradient (Fig. 2H). In contrast, mitochondria ROS scavenger (MitoTEMPO) could blunt the ROS gradient by reducing the ROS level of outer layer (Figs. 2I and EV2R). These results suggested that mitochondria ROS could promote the establishment of spheroid ROS gradient. Together, we postulated that loss of cristae remodeling proteins might enhance ROS gradient in tumor spheroid periphery by increasing mitochondria ROS production.

$H_2O_2$ is a relatively stable ROS species and capable of passing through cell membrane by selected aquaporin homologues or through free diffusion (Bienert et al, 2006), allowing it to convey signals between intra- and extracellular spaces. To figure out whether this cellular ROS gradient could result in a microenvironmental $H_2O_2$ gradient, we generated a glycosylphosphatidylinositol (GPI)-anchored $H_2O_2$ sensor (ss-HyPer7-GPI) (Fig. EV3A), in which HyPer7 can specifically detect $H_2O_2$ (Pak et al, 2020) and GPI can target the HyPer7 probe to the extracellular leaflet of cell membrane. The ss-HyPer7-GPI was mainly localized to the plasma membrane (Fig. EV3B), and the efficacy of this probe was verified by adding $H_2O_2$ or its scavenger catalase (Fig. EV3C). With this probe, we detected a peripheral extracellular $H_2O_2$ gradient (Fig. 2J), similar to the ROS gradient we detected with RoGFP. Moreover, addition of catalase could abolish this gradient (Fig. 2K).

Collectively, these results demonstrated the existence of a peripheral ROS gradient in 3D tumor spheroids that could project

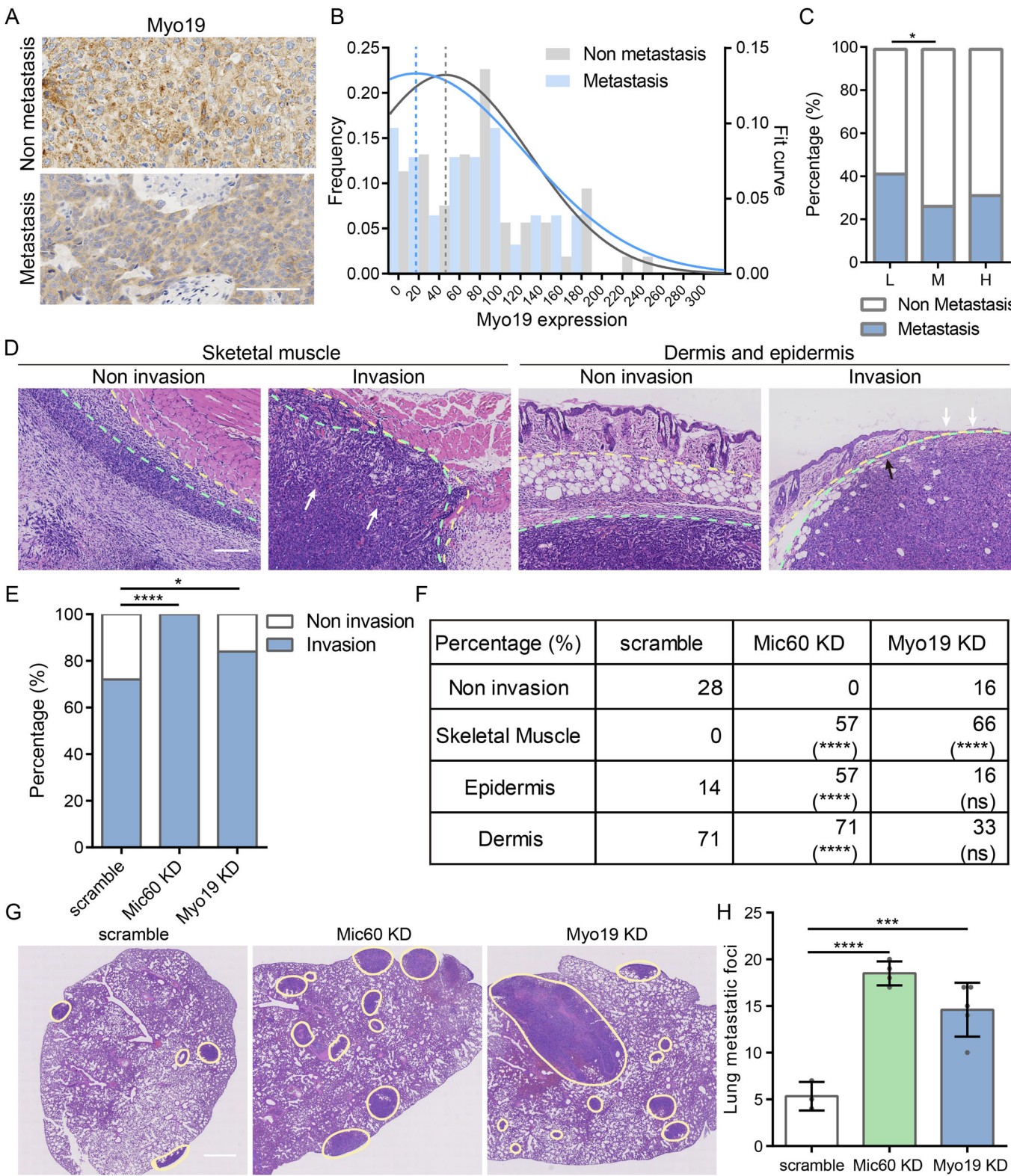

**Figure 1. Myo19 expression was negatively associated with tumor dissemination in vivo.**

(A) Representative imaging of Myo19 immunohistochemical (IHC) staining of breast carcinoma in patients with or without clinical record of lymphatic metastasis. Scale bar: 100 μm. (B) Frequency distribution histogram and Gaussian curve fitting of Myo19 IHC score in Non metastasis group and Metastasis group. Dash lines indicate peaks of the fitting curves. $N_{Non\ metastasis} = 53$, $N_{Metastasis} = 31$. (C) Percentage of metastasis incidents in Myo19 Low (L), Moderate (M) and High (H) expression groups. $N_L = 41$, $N_M = 23$, $N_H = 16$. *$p = 0.0351$. Significance was tested using Fisher's exact test. (D) Representative hematoxylin-eosin (HE) imaging of breast carcinoma tumor with or without infiltration. Yellow dash lines indicate the boundaries of skeletal muscle, or dermis and epidermis. Green dash lines indicate the boundaries of tumor tissues. Arrows: tumor infiltration into adjacent normal tissue. Scale bar: 200 μm. (E) Percentage of tumor invasion into normal tissues in 4T1 scramble, Mic60 knock down (KD) and Myo19 KD tumors. $N_{scramble} = 7$, $N_{Mic60\ KD} = 7$, $N_{Myo19\ KO} = 6$. *$p = 0.0405$; ****$p < 0.0001$. Significance was tested using Fisher's exact test. (F) Percentage of tumor infiltration into skeletal muscle, epidermis and dermis in 4T1 scramble, Mic60 KD and Myo19 KD tumors. $N_{scramble} = 7$, $N_{Mic60\ KD} = 7$, $N_{Myo19\ KO} = 6$. (G) Representative hematoxylin-eosin (HE) imaging of the lung metastatic foci. Yellow lines indicate the metastatic foci. Scale bar: 1 mm. (H) Quantification of the number of metastatic foci. Data are shown as mean ± SD. $N_{scramble} = 3$, $N_{Mic60\ KD} = 4$, $N_{Myo19\ KD} = 5$. ****$p < 0.0001$. ***$p = 0.0005$. Significance was tested using Dunnett's multiple comparisons test. Source data are available online for this figure.

into a microenvironment $H_2O_2$ gradient and can be enhanced by loss of cristae integrity.

## ROS gradient promoted spheroid invasion

Tumor dissemination begins with peripheral cells migrating and invading into surrounding stroma. To detect individual cell motion during spheroid invasion, we fluorescently labeled 5% cells in a spheroid, and monitored the trajectories of these labeled cells with live cell imaging (Fig. 3A and Movie EV1). Cells residing in the inner core only exerted slight random locomotion, while peripheral cells moved rapidly (Fig. 3A) as revealed by quantification of cell velocity along spheroid radius (Fig. 3B). We then embedded the spheroids into rat tail collagen (Fig. EV4A) to analyze 3D outward migration of spheroids generated from WT (normal ROS gradient), Myo19 KO (sharper ROS gradient) and Nrf2 KD (shallower ROS gradient) cells. We found that 3D invasion was enhanced in Myo19 KO while decreased in Nrf2 KD spheroids (Fig. EV4B,C), and the addition of catalase could inhibit the invasiveness in both WT and Myo19 KO spheroids (Figs. 3C and EV4B).

To examine whether the enhanced invasion was due to cell-autonomous factors such as the migratory ability of individual cells, or non-autonomous factors such as ROS gradient, we first analyzed 2D single cell and collective cell migration in scramble and Myo19 KD cells, but did not observe significant difference in velocity or persistence between them (Fig. EV4D–G). Furthermore, we generated spheroids with mixed WT and Myo19 KO cells labeled with different fluorescent dyes. In this way, two cell types were in the same redox microenvironment. In support of the non-autonomous factors but not different intrinsic behaviors being the major determinant for spheroid spreading, the mixed spheroid expanded in a mingled way that Myo19 KO cells did not outrun WT cells (Fig. 3D,G).

Next, we sought to test whether manipulating the ROS gradient by generating "layered spheroids" could change spheroid spreading (Fig. 3H). We seeded Myo19 KO cells around the preformed WT spheroid (WT + KO). The Myo19 KO layer corresponded to the high ROS outer layer while the preformed WT spheroid corresponded to the low ROS middle layer and high ROS inner layer (Figs. 3I and EV4H,I). We found the velocity of cells adjacent to the outer layer was higher in WT + KO spheroids compared to WT + WT spheroids (WT cells around the preformed WT spheroid) (Fig. 3L,M). Consistently, WT + KO spheroids displayed enhanced ROS gradient and outspread faster (Fig. 3J,K). Next, we switched the two cell lines (KO + WT). Compared to WT + WT spheroids with normal ROS gradient and invasion rate, KO + WT

spheroids displayed decreased ROS gradient and slowed spreading (Figs. 3J,K and EV4J,K).

Together, these results indicated that the peripheral ROS gradient could promote spheroid invasion.

## $H_2O_2$ gradient induced the chemotaxis of tumor cells

The observation that cells could migrate in response to different cellular ROS level of adjacent cells hinted that a potential ROS-related chemical cue might exist to facilitate cell motion within tumor microenvironment. To test whether extracellular $H_2O_2$ gradient could act as the chemical cue, we applied Boyden chamber transwell assay with $H_2O_2$ added in the lower chamber to create a $H_2O_2$ gradient (Fig. 4A). $H_2O_2$ levels in both upper and lower chambers were monitored to verify that the gradient could last throughout the whole experiment (Fig. 4B). With $H_2O_2$ gradient present, cell migration to the lower chamber was drastically increased, and the addition of catalase to counteract the effect of $H_2O_2$ reversed this trend (Fig. 4C,D). Moreover, addition of $H_2O_2$ in the upper chamber did not increase migratory cells into the lower chamber (Fig. EV5A–C), excluding that cell motion was directly expedited by $H_2O_2$. In fact, direct $H_2O_2$ stimulation could decrease the velocity of 2D random cell migration (Fig. EV5D).

To better monitor tumor cell chemotaxis to $H_2O_2$, and to exclude possible cell proliferation artifacts in the Boyden chamber system, we employed the μ-Slide Chemotaxis chamber combined with live cell tracking in the presence of the $H_2O_2$ gradient (Fig. 4E). Since MDA-MB-231 cells harbored higher proliferation rate, and that the proliferating cells exhibited halted migration before division and transiently peaked velocity right after cytokinesis, we used the B16-F10 cells instead. These cells showed high migration speed and metastatic potential (Nakamura et al, 2002), which were suitable for subsequent tracking and analysis. Consistent with what we observed in transwell assay, rose plot and positive forward migration index (FMI) of cell motion (Frattolin et al, 2021; Parr et al, 2019; Quast et al, 2022) revealed that cells migrated directionally towards higher concentration of $H_2O_2$ (Fig. 4F,G).

Together, these observations suggested that extracellular $H_2O_2$ gradient induced the chemotaxis of tumor cells.

## Src-RhoA signaling regulated the chemotaxis of tumor cells in $H_2O_2$ gradient

$H_2O_2$ stimulation can increase the kinase activity of purified Src protein (Heppner et al, 2018). Endogenous Src of H292 lung

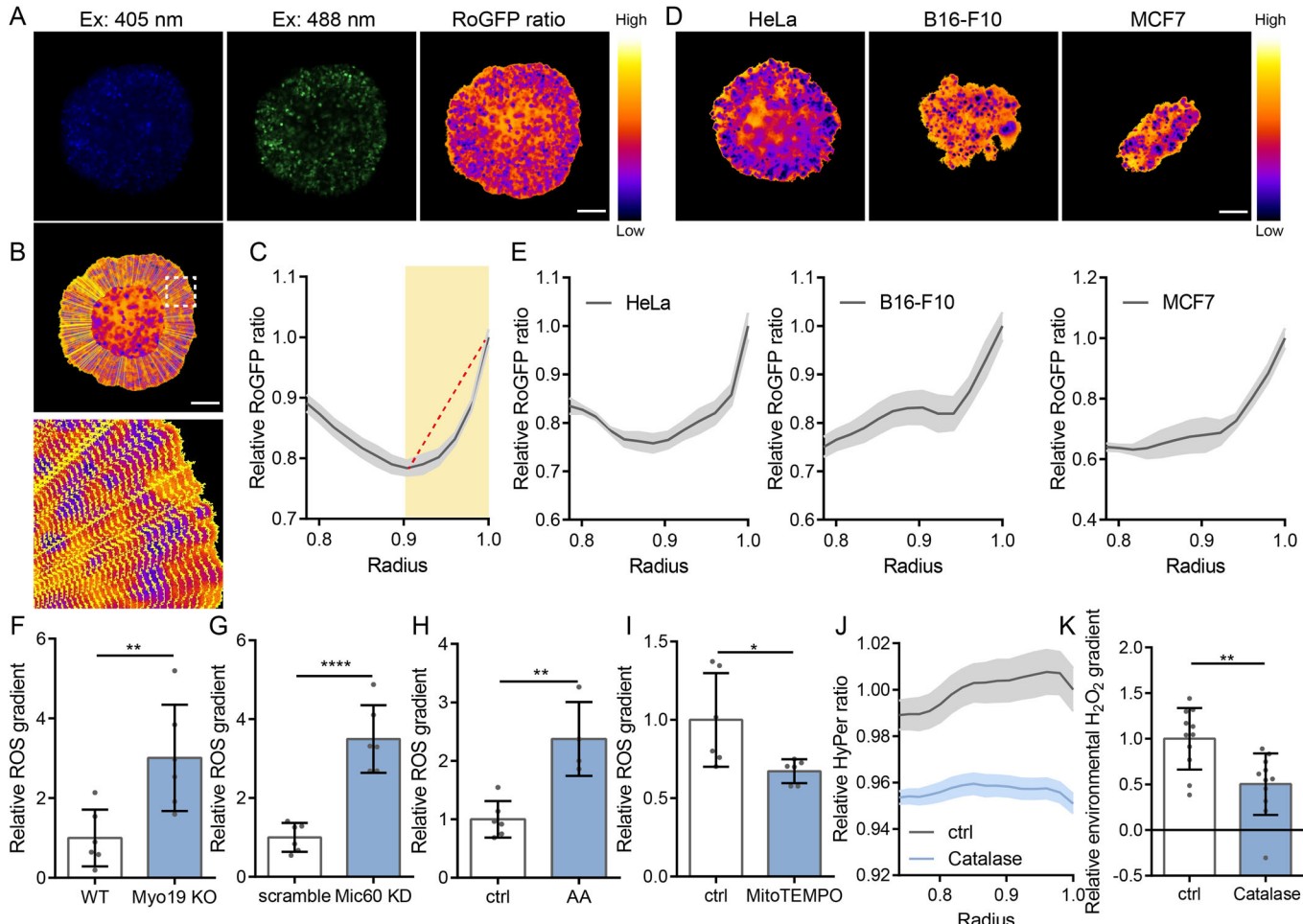

**Figure 2. Tumor spheroids displayed a peripheral microenvironmental ROS concentration gradient.**

(A) Representative fluorescent imaging of RoGFP probe in MDA-MB-231 wild type (WT) spheroids. RoGFP fluorescence was captured under the excitation wavelength of 405 nm and 488 nm, with the emission wavelength of 500–550 nm. RoGFP ratio was calculated by fluorescent intensity of Ex: 405/Ex: 488. "Ex" is short for excitation wavelength. Scale bar: 200 μm. Color bar: degree of RoGFP ratio. (B) Schematic figure showing the measuring method of RoGFP ratio. A series of inward scaling masks from the outline of the spheroid was created. Scale bar: 200 μm. (C) Quantification of RoGFP ratio along radius. The average RoGFP ratio of each concentric region in B was plotted as a function of its outer radius. Red dash line indicates the slope between the highest point in outer layer and the lowest point in middle layer, which was recognized as the ROS gradient. Data are shown as mean ± SEM. $N = 5$. (D) Representative imaging of the RoGFP ratios in HeLa, B16-F10, and MCF7 tumor spheroids expressing RoGFP probe. Scale bar: 200 μm. Color bar: degree of RoGFP ratio. (E) Quantification of the RoGFP ratios in HeLa, B16-F10 and MCF7 spheroids. Data are shown as mean ± SEM. $N_{HeLa} = 5$, $N_{B16-F10} = 4$, $N_{MCF7} = 4$. (F) Quantification of the relative ROS gradient in MDA-MB-231 WT and Myo19 knock out (KO) spheroids. Data are shown as mean ± SD. $N_{WT} = 6$, $N_{Myo19\ KO} = 6$. **$p = 0.0086$. Significance was tested using unpaired Student's $t$-test. (G) Quantification of the relative ROS gradient in WT and Mic60 KD spheroids. Data are shown as mean ± SD. $N_{WT} = 6$, $N_{Mic60\ KD} = 6$. ****$p < 0.0001$. Significance was tested using unpaired Student's $t$-test. (H) Quantification of the relative ROS gradient in spheroids treated with 1 μM Antimycin A (AA) during the whole formation process. Data are shown as mean ± SD. $N_{ctrl} = 6$, $N_{AA} = 4$. **$p = 0.0017$. Significance was tested using unpaired Student's $t$-test. (I) Quantification of the relative ROS gradient in spheroids treated with 20 μM MitoTEMPO for 3 hours. Data are shown as mean ± SD. $N_{ctrl} = 6$, $N_{MitoTEMPO} = 6$. *$p = 0.0262$. Significance was tested using unpaired Student's $t$-test. (J) Quantification of the HyPer ratio along radius in spheroids expressed with ss-HyPer7-GPI probe. Data are shown as mean ± SEM. $N_{ctrl} = 11$, $N_{Catalase} = 11$. (K) Quantification of the extracellular $H_2O_2$ gradient in spheroids treated with 2000 U/mL catalase for 5 min. Data are shown as mean ± SD. $N_{ctrl} = 11$, $N_{Catalase} = 11$. **$p = 0.0025$. Significance was tested using unpaired Student's $t$-test. Source data are available online for this figure.

mucoepidermoid carcinoma cells and NIH 3T3 embryonic fibroblast cells is also activated in response to exogenous $H_2O_2$ (Giannoni et al, 2005; Heppner et al, 2018), suggesting that Src could act as a redox sensor. By western blotting, we found $H_2O_2$ treatment increased Src Y419 phosphorylation (Fig. 4H,I), indicative of its activation in MDA-MB-231 cells. To figure out whether Src activation could regulate $H_2O_2$ chemotaxis, we generated a Src KD MDA-MB-231 cell line (Fig. EV5E) and assayed its $H_2O_2$

chemotaxis capacity using Boyden chamber transwell assay. Notably, Src KD drastically decreased cell migration towards $H_2O_2$ in the lower chamber (Fig. 4L). Similar results were also found in cells treated with PP2 and Dasatinib (Fig. 4L), both of which inhibited Src activity (Fig. EV5F).

Src can activate Rac1 to promote cell protrusion (Brugnera et al, 2002). However, inhibiting Rac1 did not hinder $H_2O_2$ induced chemotaxis although this treatment could decrease

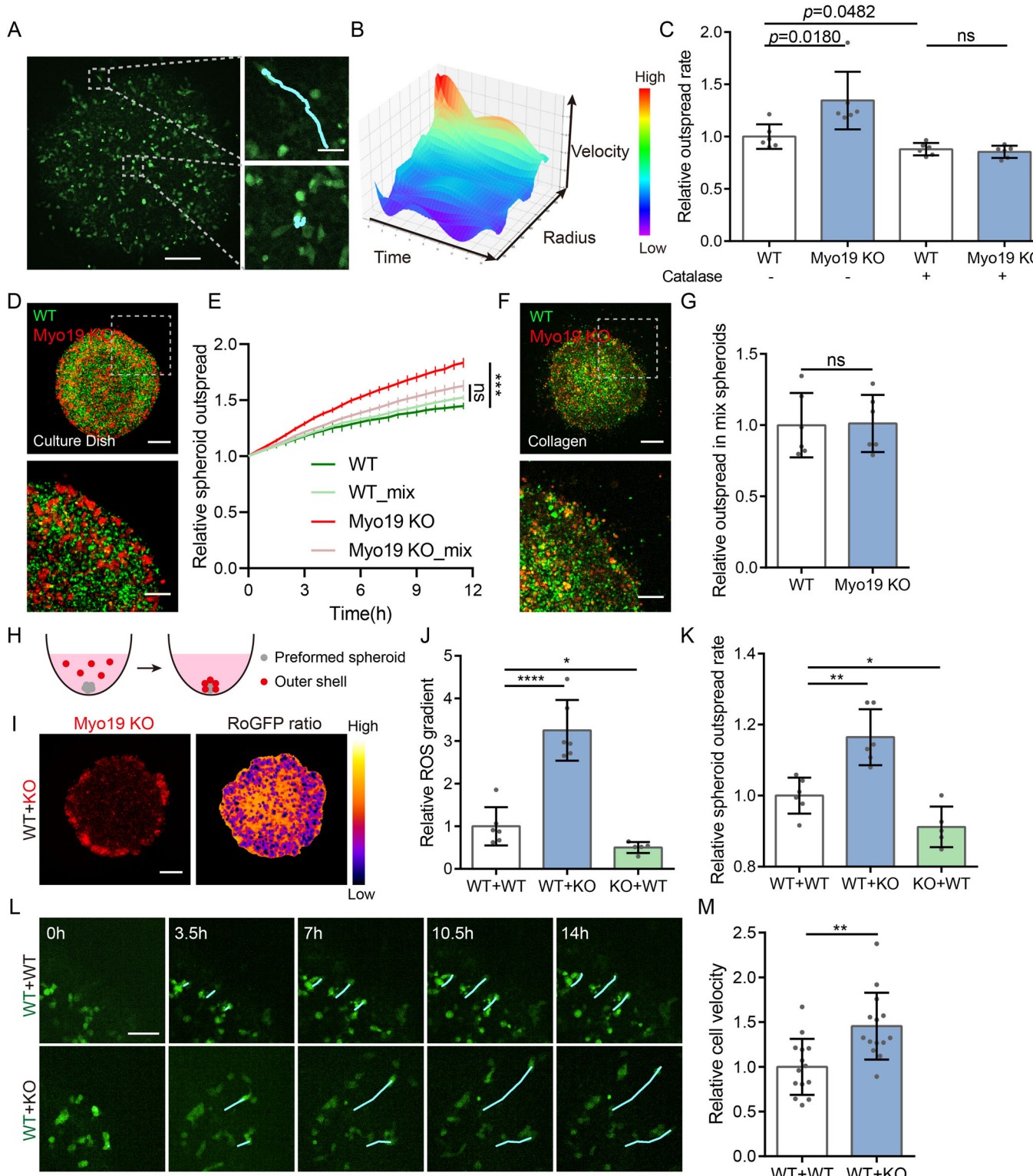

cell migration speed (Fig. 4L). Src also activates GTPase-activating proteins (GAPs) such as p190RhoGAP to inhibit RhoA (Bass et al, 2008; Huveneers and Danen, 2009), and the local inhibition of RhoA in the cell leading edge can induce polarized Myosin II activation at the cell rear to facilitate directional migration (Asokan et al, 2014). To evaluate the role of RhoA inhibition in $H_2O_2$ induced tumor cell chemotaxis, we treated MDA-MB-231 cells with RhoA activator in the transwell assay, and found that the $H_2O_2$ chemotaxis was indeed abolished (Fig. 4L). Consistently, we identified downregulation of active RhoA in

**Figure 3. ROS gradient promoted spheroid invasion.**

(A) Left: representative fluorescent imaging of spheroids after 16 h of migration on culture dish. 5% of cells were pre-stained with 1 µM CellTracker™ Green CMFDA for 1 h. Scale bar: 200 µm. Right: cropped images in the gray squares. Cyan lines indicate cell trajectories. Scale bar: 50 µm. (B) Quantification of cell velocity during spheroid invasion. Radius axis indicated the initial distance of cells from the spheroid center before migration. Color bar: degree of velocity. $N = 63$. (C) Quantification of relative outspread rate in WT and Myo19 KO spheroids treated with 5000 U/mL catalase during 48 h of invasion in 3.5 mg/mL rat tail collagen. Data are shown as mean ± SD. $N_{WT} = 6$, $N_{Myo19\ KO} = 6$, $N_{WT+Catalase} = 6$, $N_{Myo19\ KO+Catalase} = 6$. Significance was tested using unpaired Student's *t*-test. (D) Representative fluorescent imaging of WT (green)+Myo19 KO (red) mixed spheroids after 12 h of invasion on culture dish. Cells were pre-stained with 1 µM CellTracker™ Green CMFDA or 1 µM CellTracker™ Red CMTPX, and mixture of 5000 WT cells and 5000 Myo19 KO cells was used to generate tumor spheroids. Scale bar: up: 200 µm, down: 100 µm. (E) Quantification of relative outspread rate in WT, Myo19 KO, and mixed spheroid on culture dish during 12 h of invasion. Data are shown as mean ± SEM. $N_{WT} = 4$, $N_{WT\ mix} = 4$, $N_{Myo19\ KO} = 4$, $N_{Myo19\ KO\ mix} = 3$. ***$p = 0.0003$. Significance was tested using unpaired Student's *t*-test. (F) Representative fluorescent imaging of mixed spheroids after 24 h of invasion in 2 mg/mL rat tail collagen. Scale bar: up: 200 µm, down: 100 µm. (G) Quantification of relative outspread rate in WT and Myo19 KO cells of mixed spheroids during 24 hours of invasion in 2 mg/mL rat tail collagen. Data are shown as mean ± SD. $N_{WT} = 6$, $N_{Myo19\ KO} = 6$. Significance was tested using unpaired Student's *t*-test. (H) Schematic diagram showing the generation of layered spheroid. Myo19 KO cells were pre-stained with CellTracker™ Red CMTPX. 9500 WT cells were used to pre-form WT spheroid, followed by addition of 500 Myo19 KO (red) cells as the outer shell. (I) Representative fluorescent imaging of Myo19 KO and RoGFP ratio in WT + KO layered spheroids. Scale bar: 200 µm. Color bar: degree of RoGFP ratio. (J) Quantification of the relative ROS gradient in WT + WT, WT + KO and KO + WT layered spheroids. Data are shown as mean ± SD. $N_{WT+WT} = 6$, $N_{WT+KO} = 6$, $N_{KO+WT} = 6$. ****$p < 0.0001$. *$p = 0.0412$. Significance was tested using unpaired Student's *t*-test. (K) Quantification of the relative outspread rate in WT + WT, WT + KO and KO + WT layered spheroids during 24 h of invasion. Data are shown as mean ± SD. $N_{WT+WT} = 6$, $N_{WT+KO} = 6$, $N_{KO+WT} = 5$. **$p = 0.0016$. *$p = 0.0242$. Significance was tested using unpaired Student's *t*-test. (L) Representative imaging of cell motion in WT + WT and WT + KO layered spheroids during 14 h of invasion. 5% cells in the preformed WT spheroid were pre-stained with CellTracker™ Green CMFDA. Cyan lines indicate cell trajectories. Scale bar: 100 µm. (M) Quantification of relative velocity of cells adjacent to the outer shell (resided in the periphery of the preformed spheroid). $N_{WT+WT} = 14$, $N_{WT+KO} = 14$. **$p = 0.0018$. Significance was tested using unpaired Student's *t*-test. Source data are available online for this figure.

MDA-MB-231 cells upon H$_2$O$_2$ treatment by pulldown assay (Fig. 4J,K).

Together, these results suggested that the activation of Src and inhibition of RhoA regulated H$_2$O$_2$ chemotaxis in tumor cells.

Gross investigations of total or average ROS level lack the spatial resolution to reveal the role of ROS as a directional cue in solid tumor. Here, we employed RoGFP probe to investigate ROS distribution in tumor spheroids, and identified a peripheral ROS concentration gradient, which could project into an extracellular H$_2$O$_2$ gradient.

ROS, which are considered as double-edge swords, are involved in tumor initiation, proliferation, invasion and angiogenesis (Kirtonia et al, 2020). Excessive ROS would also trigger senescence and multiple cell death pathways (Circu and Aw, 2010; Kim et al, 2007; Stockwell et al, 2017). The inner core of tumor spheroids and solid tumors are characterized of various cell death pathways, namely apoptosis, necrosis and ferroptosis (Bell et al, 2001; Demuynck et al, 2020; Schafer et al, 2009; Takahashi et al, 2020). Treatment of antioxidants reduced the cell death in the inner spheroids (Schafer et al, 2009; Takahashi et al, 2020), suggesting that cell death in spheroids is associated with their redox environment.

We have previously reported that loss of Myo19 could disrupted mitochondria cristae architecture through mechanical instability (Shi et al, 2022). Disrupted cristae integrity is associated with elevated mitochondria ROS production (Majstrowicz et al, 2021), possibly through increasing electron leak during OXPHOS (Murphy, 2009). Here we showed that mitochondria ROS could contribute to the establishment of spheroid ROS gradient, which was enhanced by loss of Myo19. Apart from this, Myo19 is also reported to regulate ROS-induced mitochondria distribution to filopodia in U2OS cells (Shneyer et al, 2017), strengthening the notion that Myo19 might be involved in multiple ROS-mediated cellular processes. Other ROS-mediated effects regulated by Myo19 are of great interest for future studies.

It has been identified that low concentration of H$_2$O$_2$ can attract immune cells into injury sites (Niethammer et al, 2009). Considering the analogy of tumor to a wound that never heals

(Parisi and Vidal, 2011), the role of H$_2$O$_2$ and other possible ROS in regulating local chemotaxis of immune cells and tumor cells is worth studying. Evidence of H$_2$O$_2$ being able to trigger tumor cell migration include the elevated H$_2$O$_2$ level generated at migrating cell leading edge to enhance localized actin polymerization (Cameron et al, 2015), and that H$_2$O$_2$ treatment enhances F-actin retrograde flow in cell protrusion (Taulet et al, 2012). Efficient directional migration requires coordinated cell protrusion and retraction. Classic chemotaxis model emphasized the contractile activity of actomyosin at cell rear (Tsai et al, 2019). However, the potential role of H$_2$O$_2$ gradient in inducing polarized actomyosin contractility has not been revealed yet. Here we proposed that higher extracellular H$_2$O$_2$ at cell front could oxidatively activate Src, and further inhibited RhoA activity locally. Inhibited RhoA at cell leading edge might result in a polarized distribution of ROCK and Myosin II activation at cell rear (Keil et al, 2007; Raftopoulou and Hall, 2004), which is known to facilitate directional migration (Asokan et al, 2014; Tsai et al, 2019).

Though we have identified Src as a ROS sensor that conducted H$_2$O$_2$ chemotaxis, H$_2$O$_2$ can also modulate other protein activity and affect the chemotactic behavior of cells. The lipid phosphatase PTEN can be oxidatively inhibited by H$_2$O$_2$ (Lee et al, 2002), and PTEN can regulate cellular chemotaxis through antagonizing the PI3K signaling pathway (Kim and Dressler, 2007). H$_2$O$_2$ is also reported to increase the production of C5α-like chemotactic factor through hydrolysis of C5 in human neutrophil (Shingu and Nobunaga, 1984). Most of ROS sensors contain ROS-sensitive cysteine residues (Wang et al, 2012). Dimedone-based probes can be used to identify oxidized cysteines (Nelson et al, 2010). Recently, Yang and colleagues developed a proteomic screening approach for identifying chemoattractant-stimulated translocation of polarly distributed proteins by isolating membrane associated proteins at different stimulation time (Yang et al, 2021). Combination of this approach and mass spectrometry to identify oxidized cysteine would be helpful to determine the potential ROS sensors that regulate H$_2$O$_2$-induced chemotaxis.

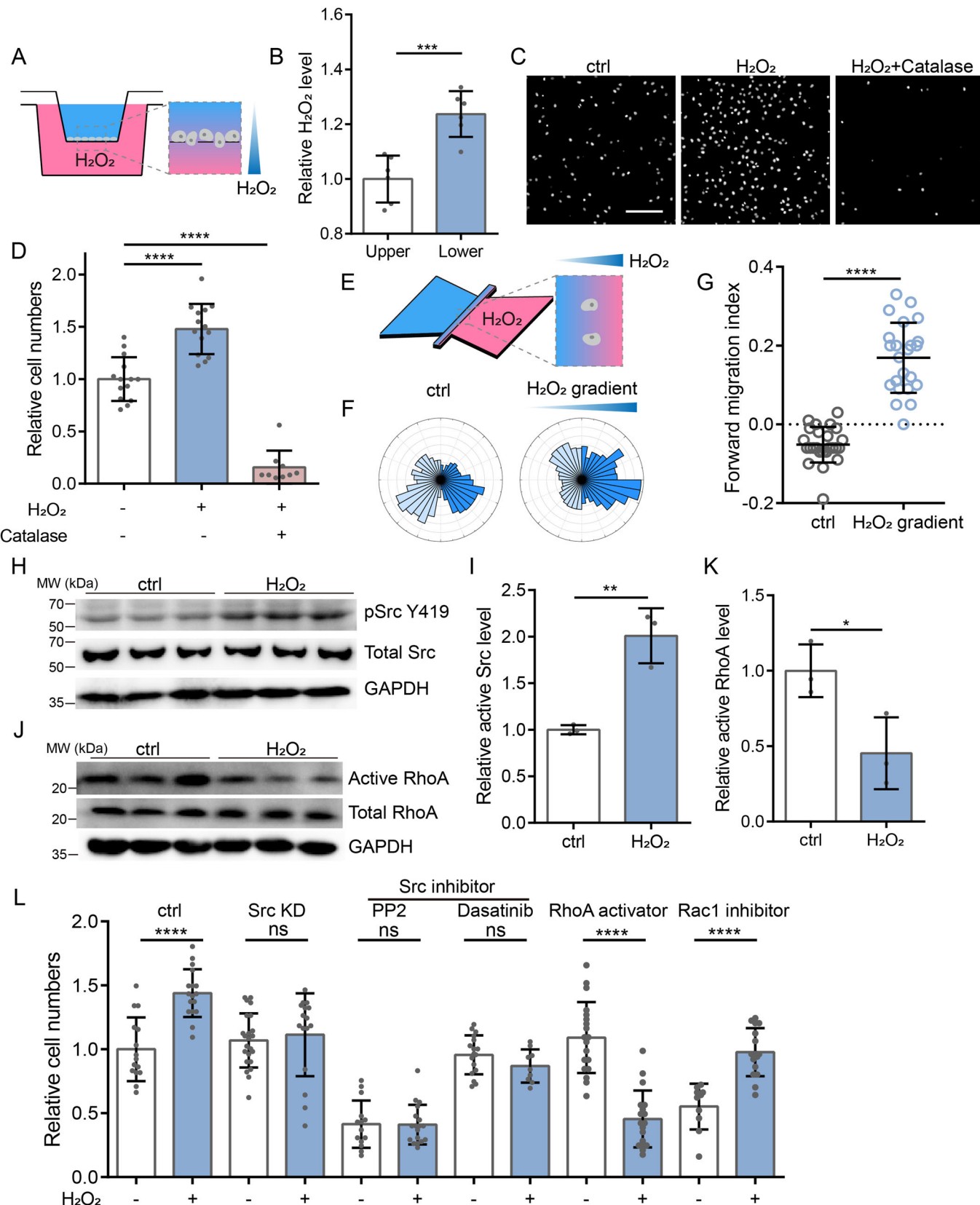

◄ **Figure 4.   H₂O₂ gradient induced chemotaxis of tumor cells through Src-RhoA signaling.**

(A) Schematic diagram showing the $H_2O_2$ chemotaxis assay using the Boyden chamber transwell assay. MDA-MB-231 cells were added in the upper chamber to allow migration. 100 µM $H_2O_2$ were added in the lower chamber to establish a $H_2O_2$ gradient. 1000 U/mL catalase was also used to scavenge $H_2O_2$ in the $H_2O_2$+Catalase group. After 12 hours of migration, cells were stained with 2.5 µg/mL Hoechst for 20 minutes and counted. (B) Quantification of relative $H_2O_2$ level in the upper and lower chamber after 12 h of cell migration. Data are shown as mean ± SD. $N_{Upper} = 6$, $N_{Lower} = 6$. ***$p = 0.0007$. Significance was tested using unpaired Student's $t$-test. (C) Representative imaging of MDA-MB-231 cells in the lower chamber. Scale bar: 200 µm. (D) Quantification of cell numbers in the lower chambers. Data are shown as mean ± SD. $N_{ctrl} = 14$, $N_{H2O2} = 14$, $N_{H2O2+Catalase} = 9$. ****$p < 0.0001$. Significance was tested using unpaired Student's $t$-test. (E) Schematic diagram showing the $H_2O_2$ chemotaxis assay using µ-Slide Chemotaxis chamber. Fresh culture medium was added in the left reservoir and 50 µM $H_2O_2$ was added in the right reservoir to establish a $H_2O_2$ gradient in the channel between the two reservoirs. B16-F10 cells were placed into the channel and their migration was monitored. (F) Rose diagrams of the trajectory of B16-F10 cell migration with or without $H_2O_2$ gradient. $N_{ctrl} = 25$, $N_{H2O2 gradient} = 23$. (G) Quantification of the forward migration index of B16-F10 cell motions. Data are shown as mean ± SD. $N_{ctrl} = 25$, $N_{H2O2 gradient} = 23$. ****$p < 0.0001$. Significance was tested using unpaired Student's $t$-test. (H) Immunoblotting of MDA-MB-231 cells treated without or with 200 µM $H_2O_2$ for 10 min. GAPDH was used as a loading control. (I) Quantification of relative phosphorylated Src Y419 level. Data are shown as mean ± SD. $N_{ctrl} = 3$. $N_{H2O2} = 3$. **$p = 0.0043$. Significance was tested using unpaired Student's $t$-test. (J) Immunoblotting of active RhoA in MDA-MB-231 cells treated without or with 200 µM $H_2O_2$ for 5 h. GAPDH was used as a loading control. (K) Quantification of relative active RhoA level. Data are shown as mean ± SD. $N_{ctrl} = 3$. $N_{H2O2} = 3$. *$p = 0.0326$. Significance was tested using unpaired Student's $t$-test. (L) Quantification of cell numbers in the lower chamber using Boyden transwell assay. 100 µM $H_2O_2$ was added in the lower chamber to establish a $H_2O_2$ gradient. In drug treatment groups, 20 µM PP2 (Src inhibitor), 10 µM Dasatinib (Src inhibitor), 2 µg/mL Rho activator II (RhoA activator) or 2 µM Ehop-016 (Rac1 inhibitor) was used. Data are shown as mean ± SD. $N_{ctrl} = 15$, $N_{ctrl+H2O2} = 17$, $N_{Src KD} = 23$, $N_{Src KD+H2O2} = 16$, $N_{PP2} = 14$, $N_{PP2+H2O2} = 18$, $N_{Dasatinib} = 17$, $N_{Dasatinib+H2O2} = 9$, $N_{Rho activator} = 19$, $N_{Rho activator+H2O2} = 18$, $N_{Ehop-016} = 10$, $N_{Ehop-016+H2O2} = 16$. ****$p < 0.0001$. Significance was tested using unpaired Student's $t$-test. Source data are available online for this figure.

# Methods

## Cell culture and DNA transfection

MDA-MB-231 cells were generously provided by Dr. Yujie Sun (Peking University). Human embryonic kidney 293T (HEK 293T) and HeLa cells were generously provided by Dr. Yuxin Yin (Peking University). 4T1 cells were generously provided by Yong Liu (Xuzhou Medical University). MCF7 cells were generously provided by Jiadong Wang (Peking University). Cells were cultured in Dulbecco's modified Eagle medium (LVN1001-1, Livning) supplemented with 10% fetal bovine serum (ST30-3302, PAN), 100 U/mL penicillin and 100 µg/mL streptomycin (CC004, MAC-GENE) at 37 °C with 5% $CO_2$. B16-F10 cells were generously provided by Fuping You (Peking University) and cultured in Roswell Park Memorial Institute (RPMI) 1640 medium (SH30809.01B, biodee) with 10% fetal bovine serum (ST30-3302, PAN), 100 U/mL penicillin and 100 µg/mL streptomycin (CC004, MACGENE) at 37 °C with 5% $CO_2$. Cells were transfected with 2 µg plasmid DNA in opti-MEM (Invitrogen) containing 2 µL Neofect™ DNA transfection reagent (TF20121201) following the protocol for 24–48 h.

## Antibodies and reagents

The following antibodies were used in this study: rabbit anti-Myo19 (HPA059715, 1:1000 for western blotting and 1:800 for IHC) and mouse anti-α-tubulin (T9026, 1:4000 for western blotting) from Sigma-Aldrich; rabbit anti-Mic60 (A2751, 1:1000 for western blotting), rabbit anti-Src (A19119, 1:1000 for western blotting) rabbit anti- pSrc Y416 (AP0452, 1:1000 for western blotting) from ABclonal; mouse anti-RhoA (ARH04, 1:800 for western blotting) from Cytoskeleton; rabbit anti-GAPDH (ab181602, 1:4000 for western blotting) from Abcam; mouse anti-Nrf2 (sc-365949, 1:800 for western blotting) from Santa Cruz; anti-mouse (sc-516102, 1:4000) and anti-rabbit (sc-2004, 1:4000) horseradish peroxidase (HRP)-conjugated secondary antibodies from Santa Cruz Biotechnology.

For reagents, CellTracker™ Red CMTPX (C34552), CellTracker™ Green CMFDA (C2925) and MitoSOX (M36008) were purchased from ThermoFisher Scientific. 2-Deoxy-D-glucose (HY-13966) was purchased from MedChemExpress. $H_2O_2$ (H112515), catalase (C163049), and MitoPY (M286913) were purchased from Aladdin. PP2 (SC1234) was purchased from Beyotime. Rho activator II (CN03) was purchased from Cytoskeleton. Dasatinib (S45672) was purchased from Yuanye. MitoTEMPO (S9733) was purchased from Selleck. Ehop-016 (BD559120) was purchased from Bidepharm. AntimycinA (ab141904) was purchased from Abcam.

## Plasmid constructions

The RoGFP probe was cloned from Cyto-RoGFP (Addgene plasmid # 49435), and was subcloned into lentiviral vector (Plvx-ac-GFP-N1) using an ABclonal MultiF Seamless Assembly kit (RK21020, ABclonal). The HyPer7 probe was cloned from pLifeAct-HyPer7 (Addgene plasmid #136464), and was subcloned into lentiviral vector (Plvx-ac-GFP-N1) with GPI-related sequences (Addgene plasmid # 182866).

## Generation of knockout and knockdown cells

The Myo19 knockout MDA-MB-231 cell was maintained by our lab and generated as previously reported (Shi et al, 2022).

The following shRNAs were cloned into pLKO.1 vector. After lentivirus infection and puromycin selection, the pooled cells were verified by the western blotting.

scramble (human):
5′-AACGCTGCTTCTTCTTATTTA-3′;
Mic60 (human):
5′-GTCTAGAAATGAGCAGGTTTA-3′
scramble (mouse):
5′-AACGCTGCTTCTTCTTATTTA-3′
Myo19 (mouse):
5′-CCAGAACTTCATAGAAAGATA-3′

## Spheroid formation assay

Spheroids were generated using round-bottom ultra-low-attachment 96 wells (7007, Corning) as previously reported (Vinci et al, 2012). Cells were washed with DPBS (Dulbecco's phosphate-buffered saline,

containing no $Ca^{2+}$ and $Mg^{2+}$), trypsinized and centrifuged at 1000 rpm for 3 min followed by resuspension with fresh DMEM. Resuspended cells were counted and $10^4$ cells in 100 μL culture media were added into each well. For mixed spheroid assay, cells were pre-stained with 1 μM CellTracker™ Red CMTPX or CellTracker™ Green CMFDA for 1 hour, and mixture of 5000 WT cells and 5000 Myo19 KO cells was added into each well. For layered spheroid assay, 9500 cells added into each well. After 2 h of sedimentation to establish preformed spheroids, another 500 cells were added. After 2 days, spheroids were collected and washed with DPBS to prepare for the following experiments.

## Spheroid collagen invasion assay

For collagen invasion assays, well prepared spheroids were embedded into 3.5 mg/mL collagen type I (354249, Corning) in 12-well culture plates to allow invasion for indicated time. Bright-field images were acquired by Olympus IX83.

## Spheroid RoGFP ratio and HyPer ratio analysis

Cells transfected with RoGFP or ss-HyPer7-GPI probe were used to generate spheroids. Fluorescence of spheroids was acquired by spinning disk confocal microscope (Dragonfly imaging system), with the excitation wavelength of 405 nm or 488 nm, and the emission wavelength of 500–550 nm. RoGFP ratio was calculated as the fluorescent intensity under the excitation wavelength of 405 nm divided by the intensity excitation wavelength of 488 nm (Ex: 405 nm/Ex: 488 nm) using ImageJ software. Similarly, HyPer ratio was calculated as Ex: 488 nm/Ex: 405 nm.

To quantify the ROS gradient or extracellular $H_2O_2$ gradient, a series of inward scaling masks from the outline of each spheroid were created using the ROI manager (ImageJ). The average RoGFP ratio of each concentric region was plotted as a function of its outer radius, and the slope of the line connecting the highest point in outer layer and the lowest point in middle layer was then calculated as the gradient.

## Western blotting

For western blotting, cells were washed with DPBS twice and lysed with appropriate volumes of RIPA buffer (50 mM Tris-HCl, pH 8.0, 150 mM NaCl, 1% Triton X-100, 0.5% Na-deoxycholate, 0.1% SDS, 1 mM EDTA and protease inhibitor cocktail) for 10 min on ice. The cell lysis was centrifuged at 13,572 g for 10 minutes at 4 °C and the supernatants were collected. Then, 5× SDS loading buffer was added to the supernatants and the mixtures were boiled for 10 minutes at 95 °C. Protein samples were separated on 10-15% SDS-PAGE gels and transferred onto nitrocellulose filter membranes by wet electrophoretic transfer, followed by first antibodies incubation at 4 °C overnight or room temperature for 2 hours, and second antibodies incubation at room temperature for 1 hour. The X-ray films were used to detect and record the band intensities. The fixed X-ray films were scanned to obtain digital images. The images were processed by ImageJ software.

## Active RhoA Pulldown Assay

Cells were lysed with IP lysis buffer (150 mM NaCl, 25 mM Tris–HCl, 0.5% NP-40, pH 7.4 and protease inhibitor cocktail) for 30 min with gentle rotating. The cell lysis was centrifuged at 13,572 g for 10 minutes to remove the insoluble components, and the supernatants were collected. 20 μg Rhotekin-RBD Beads (RT02, Cytoskeleton) were washed three times using IP lysis buffer. Cell lysates (800 μg) were incubated with the beads at 4 °C for 1 h. After incubation, the beads were washed three times using the IP lysis buffer. The 1x loading buffer was added to suspend the beads and boiled for 10 min at 95 °C. Active RhoA was then detected by western blot using anti-RhoA (ARH04, Cytoskeleton) antibody.

## Boyden chamber migration assay

MDA-MB-231 cells were washed with DPBS, trypsinized and centrifuged at 1000 rpm for 3 min followed by resuspension with fresh DMEM. Cells were counted and 250 μL cell suspension with the concentration of $5 \times 10^5$ cells/mL was added onto the upper chamber of Boyden chamber (3422, Corning). After 5 minutes, 1 mL culture medium with or without 100 μM $H_2O_2$ or 1000 U/mL catalase was slowly added in the lower chamber. After 12 hours of migration, cells were stained with 2.5 μg/mL Hoechst for 20 minutes and washed with DPBS. Cells in the upper chamber were gently wiped off and the migratory cells were captured and counted. To measure the persistency of ROS gradient, cells were cultured in DMEM without phenol red (CC012, MACGENE). After 12 hours of migration, culture medium in the upper chamber and the lower chamber was collected separately and $H_2O_2$ level was measured using Hydrogen Peroxide Assay Kit (S0038, Beyotime).

## Chemotactic chamber assay

2D chemotaxis assay was performed using μ-Slide Chemotaxis (80326, ibidi GmbH, Martinsried, Germany) as instructed. B16-F10 cells were washed with DPBS, trypsinized and centrifuged at 1000 rpm for 3 min followed by resuspension with fresh culture medium. Resuspended cells were placed into the channel between the two reservoirs. After attachment, the reservoirs were filled with $CO_2$-independent media (L1518, Sigma-Adrich) with or without chemoattractant (50 μM $H_2O_2$). Cell motions were captured every 20 min for 9 h. Images were analyzed using "Manual tracking" and "Chemotaxis tool" plugin of ImageJ.

## Mammary fad pad injection of breast cancer cells

Mammary fad pad injection was performed as previously reported (Yu et al, 2017). In general, 4T1 cells were washed with DPBS, trypsinized, and centrifuged at 1000 rpm for 3 min followed by resuspension with fresh DPBS. Cells were counted and $10^6$ cells in 100 μL DPBS were injected into the mammary fat pad of the forth nipple of Balb/c female mice with an insulin syringe. Successful injection was confirmed by checking for swelling of the fat pad.

To examine the tumor infiltration, two weeks after the implantation of 4T1 scramble, Mic60 KD and Myo19 KD cells, mice were sacrificed and tumors with adjacent tissues were taken out and fixed by 4% paraformaldehyde (BL539A, biosharp) for 24 hours. Samples were then embedded in paraffin and sliced followed by hematoxylin-eosin (HE) staining. All animal experiments were performed in accordance with relevant guidelines and regulations and were approved by the Animal Care and Use Committee of Peking University Third Hospital.

## Breast carcinoma samples

Primary breast carcinoma samples and clinical records of patients were obtained from Peking University Third Hospital. Breast carcinoma samples were fixed by 4% paraformaldehyde, embedded in paraffin and sliced followed by standard immunohistochemistry (IHC) protocol to examine the Myo19 expression. Myo19 IHC score was calculated as the positive intensity multiplied percentage of positive cells. The procedure was approved by the Medical Science Research Ethics Committee of Peking University Third Hospital. Informed consent was obtained from all subjects and the experiments were conformed to the principles set out in the WMA Declaration of Helsinki and the Department of Health and Human Services Belmont Report.

## Statistics

The number of biological and technical replicates and the number of samples are indicated in figure legends, the main text and the Materials and Methods section. Data are mean ± SD or mean ± SEM as indicated in the figure legends and Expanded View figure legends. "Relative level" was calculated to normalize the experimental values. To do this, we first calculated the average level of control group, and then divided the experimental values in each group with this average value.

# Data availability

This study includes no data deposited in external repositories. Data supporting the findings of this work are available within the paper and its supplementary files.

# Peer review information

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

## Acknowledgements

We thank Xin Yi, Fang Zhou, Chunlei Zhang, Zifeng Zhen from Peking University for helpful discussion. We thank the Pathology Department at Peking University Third Hospital, for assistance with HE and IHC staining of tumor samples. This work was supported by funding from the National Key R&D Program of China (2022YFC3401100) and the National Natural Science Foundation of China 32122029 for Congying Wu, 32300635 for Peng Shi; China Postdoctoral Science Foundation (2022M710263) for Peng Shi. Peng Shi was supported in part by the Postdoctoral Fellowship of Peking-Tsinghua Center for Life Sciences.

## Author contributions

**Xiaoyu Ren**: Conceptualization; Data curation; Formal analysis; Validation; Investigation; Visualization; Methodology; Writing—original draft; Writing—review and editing. **Peng Shi**: Conceptualization; Resources; Data curation; Formal analysis; Funding acquisition; Investigation; Writing—original draft; Writing—review and editing. **Jing Su**: Resources; Data curation; Investigation; Methodology. **Tonghua Wei**: Conceptualization; Data curation; Investigation. **Jiayi Li**: Conceptualization; Investigation; Methodology. **Yiping Hu**: Conceptualization; Data curation; Investigation. **Congying Wu**: Conceptualization; Resources; Supervision; Funding acquisition; Validation; Visualization; Writing—original draft; Project administration; Writing—review and editing.

## Disclosure and competing interests statement

The authors declare no competing interests.

# Expanded View Figures

**Figure EV1.  Cristae sculpting protein Myo19 and Mic60 promoted tumor invasion and metastasis without affecting tumor growth.**

(**A**) Myo19 expression in tumors compared to normal tissues in EMBL-EBI (EMBL's European Bioinformatics Institute) database. (**B**) Quantification of Myo19 IHC score and Ki-67 index. Pearson correlation analysis showed no significance. Pearson $r = -0.1092$, $p = 0.3319$. $N = 81$. (**C**) Representative imaging of Ki-67 IHC staining of breast carcinoma in Myo19 Moderate (M) and Low (L) expression group. Scale bar: 100 μm. (**D**) Quantification of Ki-67 index in Myo19 Low (L), Moderate (M) and High (H) expression groups. Data are shown as mean ± SD. $N_L = 41$, $N_M = 23$, $N_H = 16$. Significance was tested using Fisher's exact test. (**E**) Representative image of 4T1 scramble, Mic60 KD and Myo19 KD solid tumors. (**F**) Quantification of the tumor weight of 4T1 scramble, Mic60 KD and Myo19 KD solid tumors. Data are shown as mean ± SD. $N_{scramble} = 7$, $N_{Mic60\ KD} = 7$, $N_{Myo19\ KD} = 7$. Significance was tested using unpaired Student's $t$-test. (**G**) Quantification of the maximal diameter of 4T1 scramble, Mic60 KD and Myo19 KD solid tumors. Data are shown as mean ± SD. $N_{scramble} = 7$, $N_{Mic60\ KD} = 7$, $N_{Myo19\ KD} = 7$. Significance was tested using unpaired Student's $t$-test. (**H**) Representative electron microscope (EM) images of MDA-MB-231 wild type (WT), Myo19 KO and Mic60 KD cells. Scale bar: 1 μm. Zoomed image scale bar: 300 nm. Source data are available online for this figure.

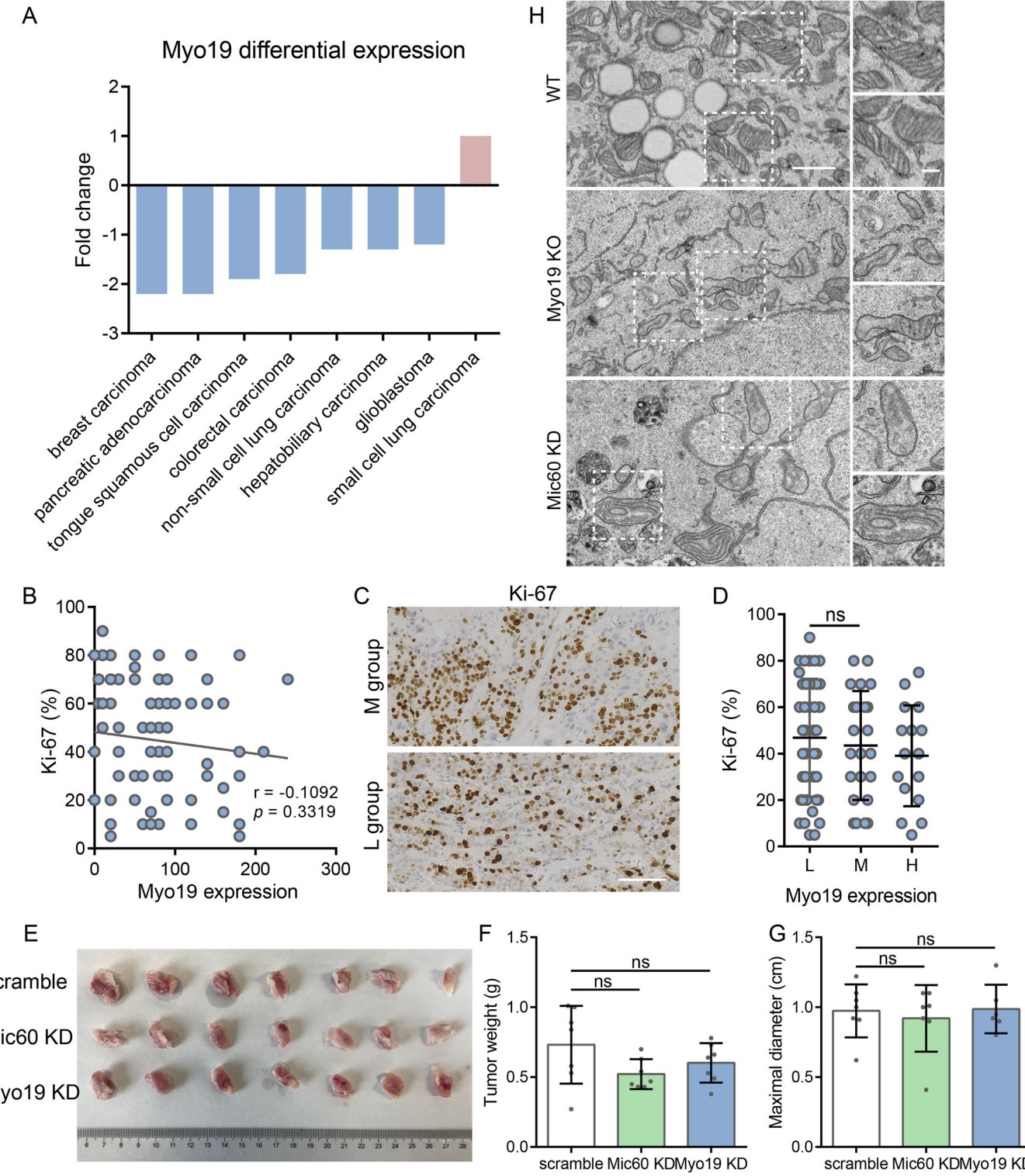

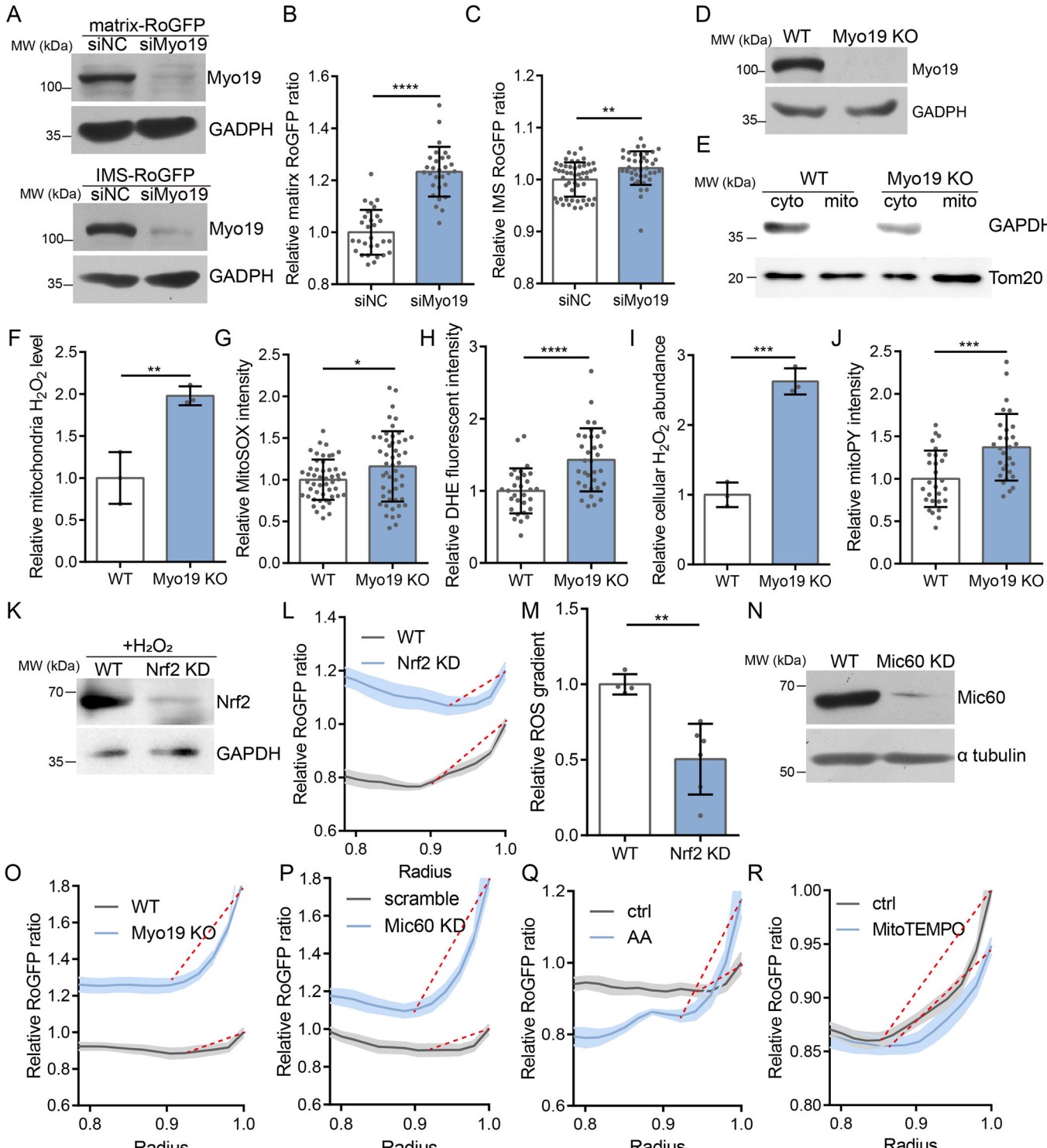

Figure EV2.    Loss of Myo19 promoted mitochondria ROS production and spheroid ROS gradient.

(A) Immunoblotting of mitochondria matrix- and intermembrane space (IMS)-RoGFP expressed MDA-MB-231 cells with siMyo19. GAPDH was used as a loading control. (B) Quantification of the RoGFP ratio in matrix-RoGFP expressed cells with siMyo19. Data are shown as mean ± SD. $N_{siNC} = 29$, $N_{siMyo19} = 29$. ****, $p < 0.0001$. Significance was tested using unpaired Student's $t$-test. (C) Quantification of the RoGFP ratio in IMS-RoGFP expressed cells with siMyo19. Data are shown as mean ± SD. $N_{siNC} = 51$, $N_{siMyo19} = 40$. **, $p = 0.0020$. Significance was tested using unpaired Student's $t$-test. (D) Immunoblotting of WT and Myo19 KO cells. GAPDH was used as a loading control. (E) Immunoblotting of isolated mitochondria from MDA-MB-231 WT and Myo19 KO cells. GAPDH was used to indicate cytosolic proteins and Tom 20 was used to indicate mitochondria proteins. (F) Quantification of relative $H_2O_2$ level of isolated mitochondria. Data are shown as mean ± SD. $N_{WT} = 3$, $N_{Myo19\ KO} = 3$. **, $p = 0.0067$. Significance was tested using unpaired Student's $t$-test. (G) Quantification of relative MitoSOX intensity in MDA-MB-231 WT and Myo19 KO cells. Cells were treated with 500 nM MitoSOX for 30 minutes. Data are shown as mean ± SD. $N_{WT} = 49$, $N_{Myo19\ KO} = 50$. *, $p = 0.0233$. Significance was tested using unpaired Student's $t$-test. (H) Quantification of relative DHE intensity of MDA-MB-231 WT and Myo19 KO cells. Cells were treated with 10 μM DHE for 20 minutes. Data are shown as mean ± SD. $N_{WT} = 30$, $N_{Myo19\ KO} = 33$. ****$p < 0.0001$. Significance was tested using unpaired Student's $t$-test. (I) Quantification of relative $H_2O_2$ level in WT and Myo19 KO cell lysis. Data are shown as mean ± SD. $N_{WT} = 3$, $N_{Myo19\ KO} = 3$. ***$p = 0.0004$. Significance was tested using unpaired Student's $t$-test. (J) Quantification of relative mitoPY intensity in MDA-MB-231 WT and Myo19 KO cells. Cells were treated with 20 μM mitoPY for 3 h. Data are shown as mean ± SD. $N_{WT} = 28$, $N_{Myo19\ KO} = 29$. ***$p = 0.0003$. Significance was tested using unpaired Student's $t$-test. (K) Immunoblotting of MDA-MB-231 WT and Nrf2 KD cells treated with 100 μM $H_2O_2$ for 1 h. GAPDH was used as a loading control. (L) Quantification of the relative RoGFP ratio along radius in WT and Nrf2 KD spheroids. Red dash line indicates ROS gradient. Data are shown as mean ± SEM. $N_{WT} = 4$, $N_{Nrf2\ KD} = 6$. (M) Quantification of the relative ROS gradient in WT and Nrf2 KD spheroids. Data are shown as mean ± SD. $N_{WT} = 4$, $N_{Nrf2\ KD} = 6$. **$p = 0.0037$. Significance was tested using unpaired Student's $t$-test. (N) Immunoblotting of WT and Mic60 KD cells. α tubulin was used as a loading control. (O) Quantification of the relative RoGFP ratio along radius in WT and Myo19 KO spheroids. Red dash line indicates ROS gradient. Data are shown as mean ± SEM. $N_{WT} = 6$, $N_{Myo19\ KO} = 6$. (P) Quantification of the relative RoGFP ratio along radius in WT and Mic60 KD spheroids. Red dash line indicates ROS gradient. Data are shown as mean ± SEM. $N_{WT} = 6$, $N_{Mic60\ KD} = 5$. (Q) Quantification of the relative RoGFP ratio along radius in spheroids treated with 1 μM Antimycin A (AA) during the whole formation process. Red dash line indicates ROS gradient. Data are shown as mean ± SEM. $N_{ctrl} = 6$, $N_{AA} = 4$. (R) Quantification of the relative RoGFP ratio along radius in spheroids treated with 20 μM MitoTEMPO for 3 hours. Red dash line indicates ROS gradient. Data are shown as mean ± SEM. $N_{ctrl} = 6$, $N_{MitoTEMPO} = 6$. Source data are available online for this figure.

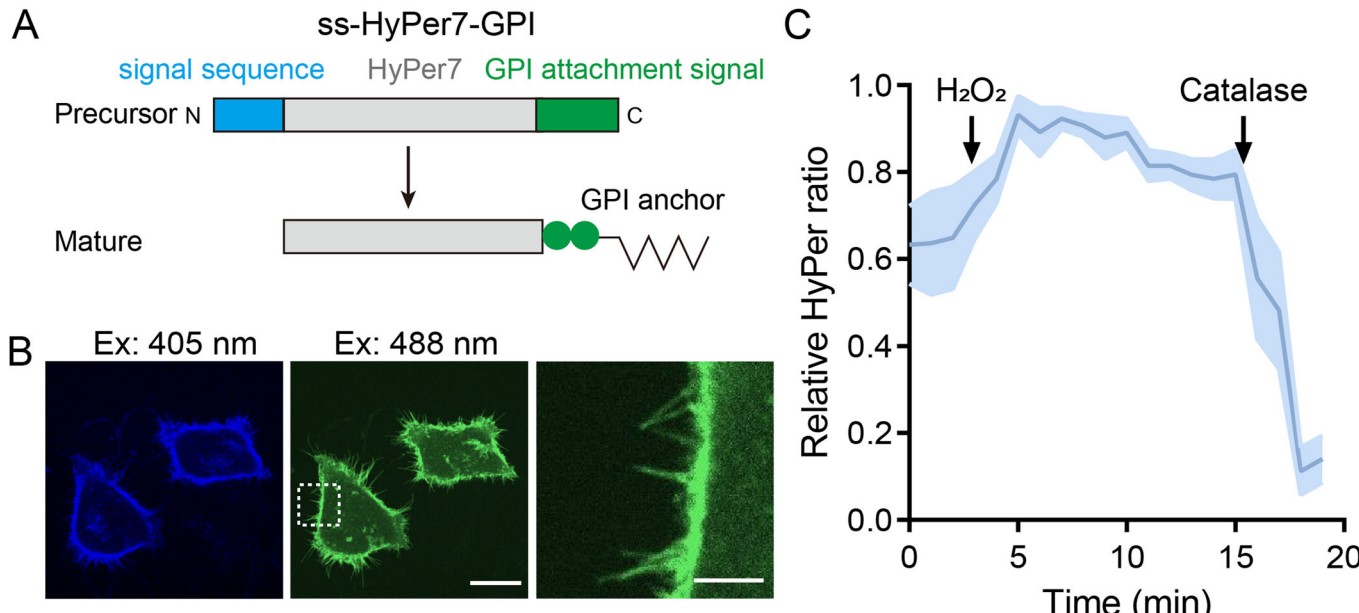

**Figure EV3.  ss-HyPer7-GPI probe indicated extracellular H₂O₂ level.**

(A) Schematic diagram showing the construct of ss-HyPer7-GPI probe. The HyPer7 sensor was fused with N-terminal endoplasmic reticulum (ER)-targeting signal sequence (ss), and C-terminal GPI-attachment signal. In the mature probe, these two sequences of the precursor probe were removed and GPI was attached to HyPer7. (B) Localization of the HyPer7 fluorescence in MDA-MB-231 cells expressing ss-HyPer7-GPI. Scale bar: 20 μm. Scale bar of the cropped images: 10 μm. (C) Quantification of the relative HyPer ratio in MDA-MB-231 cells expressing ss-HyPer7-GPI. 150 μM H₂O₂ and 2000 U/mL catalase were added in the culture medium at the indicated time. $N = 7$. Source data are available online for this figure.

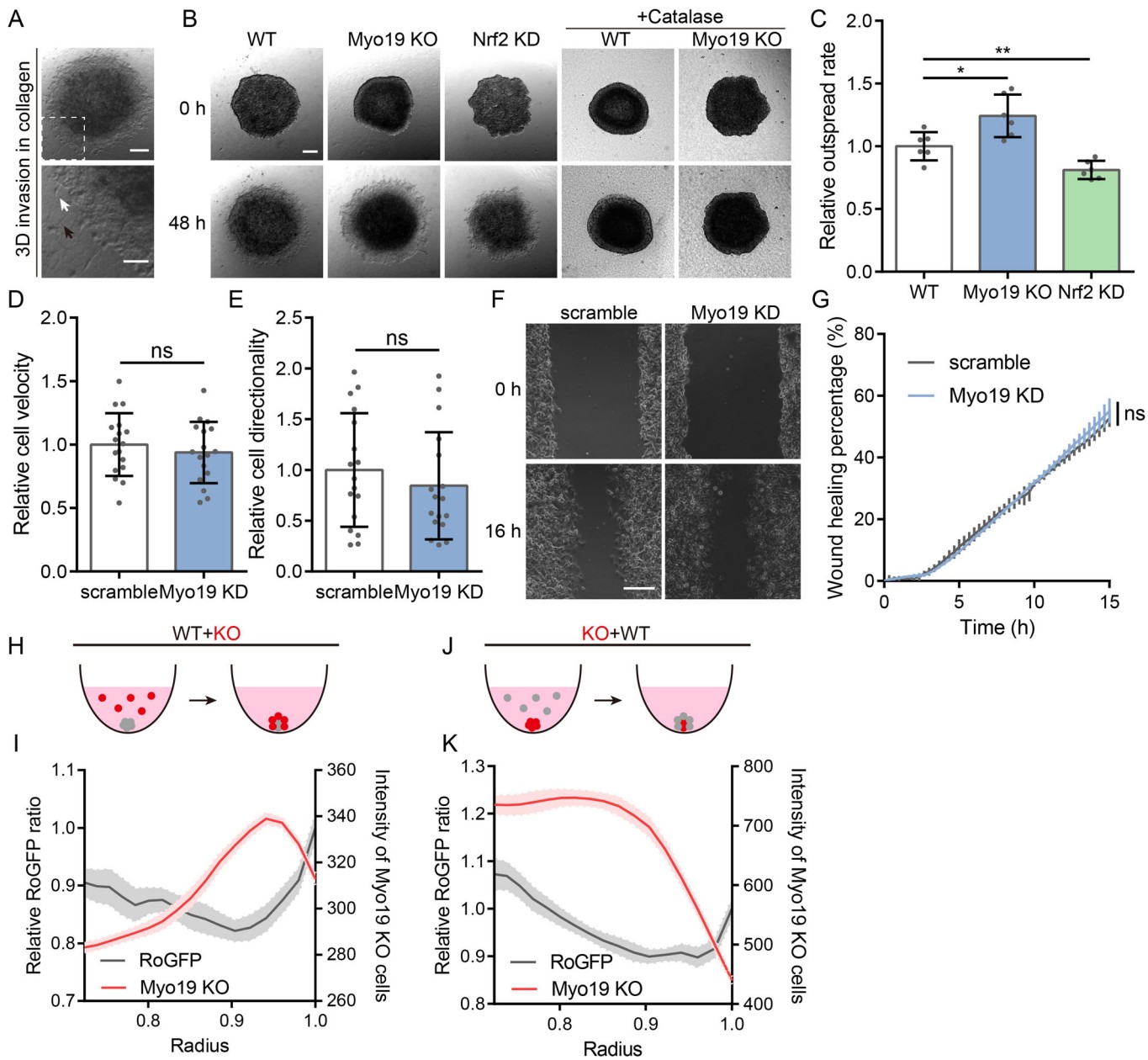

**Figure EV4. Loss of Myo19 promoted 3D spheroid invasion but not migratory ability.**

(A) Up: representative imaging of MDA-MB-231 spheroid invasion in 3.5 mg/mL rat tail collagen. Scale bar: 200 μm. Down: the cropped image in the white square. White arrow indicates the invading cell. Black arrow indicates the collagen fiber. Scale bar: 100 μm. (B) Representative images of spheroid invasion in 3.5 mg/mL rat tail collagen. 5000 U/mL catalase was added during 48 hours of invasion. Scale bar: 200 μm. (C) Quantification of relative outspread rate in WT, Myo19 KO and Nrf2 KD spheroids during 48 hours of invasion. Data are shown as mean ± SD. $N_{WT} = 6$, $N_{Myo19\ KO} = 6$, $N_{Nrf2\ KD} = 6$. *$p = 0.0154$. **$p = 0.0061$. Significance was tested using unpaired Student's *t*-test. (D) Quantification of the velocity of B16-F10 cell random motion. Data are shown as mean ± SD. $N_{scramble} = 17$, $N_{Myo19\ KD} = 17$. Significance was tested using unpaired Student's *t*-test. (E) Quantification of the directionality of B16-F10 cell random motion. Data are shown as mean ± SD. $N_{scramble} = 17$, $N_{Myo19\ KD} = 17$. Significance was tested using unpaired Student's *t*-test. (F) Representative imaging of wound scratch assay of B16-F10 scramble and Myo19 KD cells. Scale bar: 200 μm. (G) Quantification of percentage of wound healing within 16 hours. Data are shown as mean ± SD. $N_{scramble} = 5$. $N_{Myo19\ KD} = 5$. Significance was tested using unpaired Student's *t*-test. (H) Schematic diagram showing the generation of WT + KO layered spheroid. Myo19 KO cells were pre-stained with CellTracker™ Red CMTPX. 9500 WT cells were used to pre-form WT spheroid, followed by addition of 500 Myo19 KO (red) cells as the outer shell. (I) Quantification of the relative RoGFP ratio and the fluorescent intensity of cell dye-stained Myo19 KO cells in WT + KO spheroids. Myo19 KO cells were pre-stained with CellTracker™ Red CMTPX. Data are shown as mean ± SEM. $N = 6$. (J) Schematic diagram showing the generation of KO + WT layered spheroid. Myo19 KO cells were pre-stained with CellTracker™ Red CMTPX. 9500 Myo19 KO (red) cells were used to pre-form Myo19 KO spheroid, followed by addition of 500 WT cells as the outer shell. (K) Quantification of the relative RoGFP ratio and the fluorescent intensity of cell dye-stained Myo19 KO cells in KO + WT spheroids. Myo19 KO cells were pre-stained with CellTracker™ Red CMTPX. Data are shown as mean ± SEM. $N = 6$. Source data are available online for this figure.

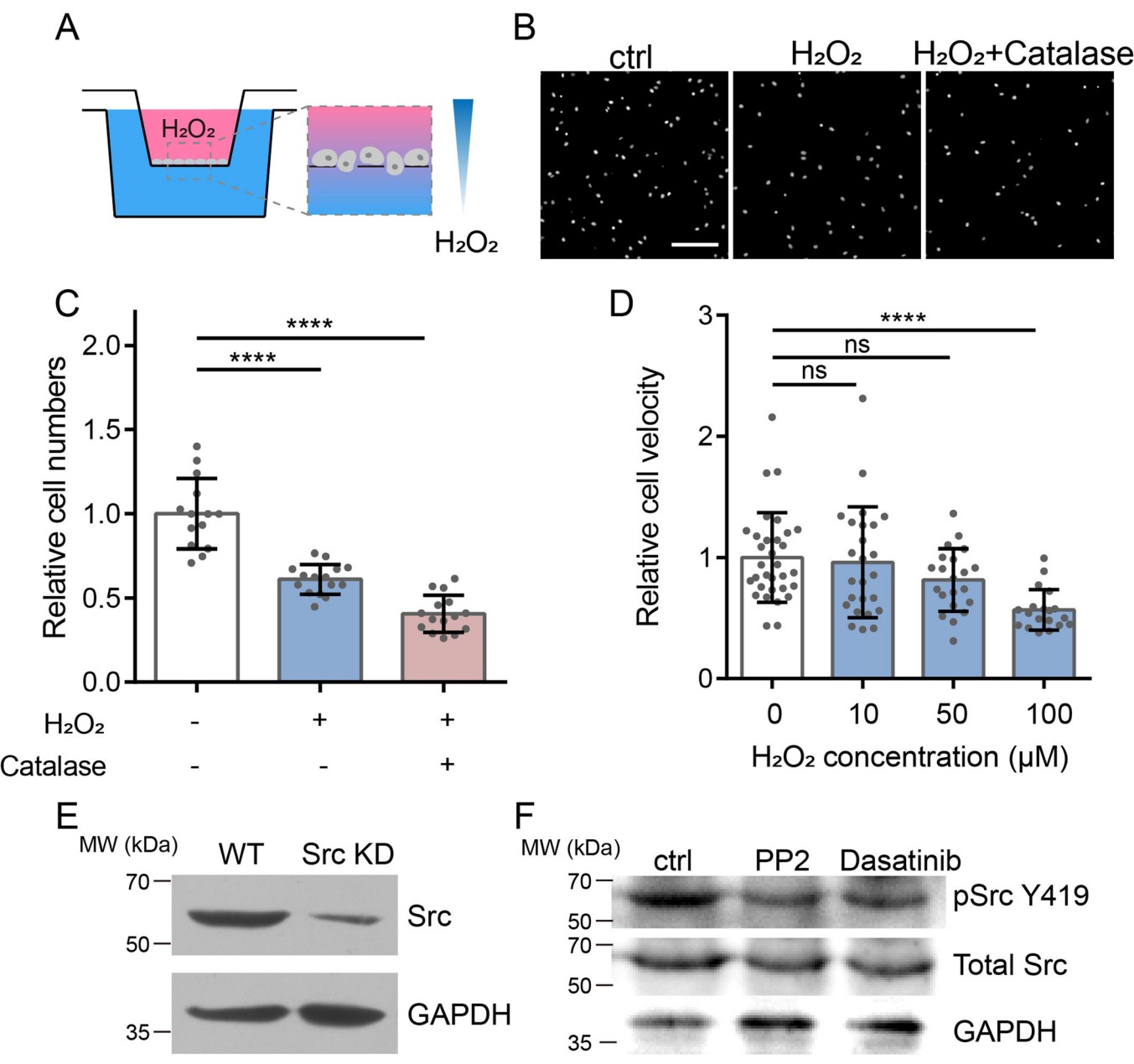

**Figure EV5.  Direct H₂O₂ stimulation inhibited the velocity of cell migration.**

(A) Schematic diagram showing the reverse $H_2O_2$ chemotaxis assay using the Boyden chamber transwell assay. MDA-MB-231 cells and 100 μM $H_2O_2$ were added in the upper chamber. 1000 U/mL catalase was also used to scavenge $H_2O_2$ in the $H_2O_2$+Catalase group. After 12 hours of migration, cells were stained with 2.5 μg/mL Hoechst for 20 min and counted. (B) Representative imaging of cells in the lower chamber. Scale bar: 200 μm. (C) Quantification of cell numbers in the lower chambers. Data are shown as mean ± SD. $N_{ctrl} = 14$, $N_{H2O2} = 15$, $N_{H2O2+catalase} = 15$. ****$p < 0.0001$. Significance was tested using unpaired Student's $t$-test. (D) Quantification of the relative velocity of B16-F10 random cell motion. Cells were treated with $H_2O_2$ of indicated concentration. Images were captured for 10 hours at the interval of 20 minutes. Data are shown as mean ± SD. $N_{0\ \mu M} = 32$, $N_{10\ \mu M} = 25$, $N_{50\ \mu M} = 21$, $N_{100\ \mu M} = 19$, ****$p < 0.0001$. Significance was tested using unpaired Student's $t$-test. (E) Immunoblotting of MDA-MB-231 WT and Src KD cells. GAPDH was used as a loading control. (F) Immunoblotting of MDA-MB-231 cells treated with 20 μM PP2 or 10 μM Dasatinib for 12 h. GAPDH was used as a loading control. Source data are available online for this figure.

