## [Peer Review File · EMBO Reports]

Loss of Myo19 increases metastasis by enhancing microenvironmental ROS gradient and chemotaxis

Xiaoyu Ren, Peng Shi, Jing Su, Tonghua Wei, Jiayi Li, Yiping Hu, and Congying Wu
DOI: 10.15252/embr.202357624

Corresponding author(s): Congying Wu (congyingwu@hsc.pku.edu.cn) , Peng Shi (peng@bjmu.edu.cn)

Review Timeline:

Submission Date:	10th Jun 23
Editorial Decision:	16th Jun 23
Appeal Received:	23rd Jun 23
Editorial Decision:	31st Jul 23
Revision Received:	26th Oct 23
Editorial Decision:	7th Dec 23
Revision Received:	10th Dec 23
Accepted:	19th Dec 23

Editor: Deniz Senyilmaz Tiebe

Transaction Report:

Dear Prof. Wu,

Thank you for submitting your manuscript to EMBO Reports. I have read your study carefully and discussed it with the other members of our editorial team including our chief editor Dr. Bernd Pulverer. I regret to inform you that we have decided not to pursue publication of this manuscript in its current form, but we would be happy to reconsider it with some additional analysis as mentioned below.

I apologize for this unusual delay in getting back to you, which was caused by the current high rate of new submissions to our office, affecting our usually much shorter editorial handling time.

We appreciate your findings supporting that loss of Myo19 in cancer enhances the H₂O₂ gradient, thereby promoting invasive behavior of cancer cells. We realize that these findings are as such of interest to the field. However, we also note that, in our view, whether the changes in H₂O₂ gradient is necessary for the Myo19 depletion induced spheroid invasion has not been conclusively demonstrated - e.g. by comparing spheroid invasion of WT and KO spheroids in the presence of catalase. We feel that these points would come up during peer-review as well. As such, we concluded that the advance provided is not sufficient for publication in EMBO Reports in the current form of the manuscript. That said, we would be happy to send the manuscript out for formal peer-review should you be willing to include additional data addressing these concerns.

Thank you for giving us the opportunity to consider this manuscript.

Yours sincerely,

Deniz Senyilmaz Tiebe, PhD
Scientific Editor
EMBO Reports

** As a service to authors, EMBO Press provides authors with the ability to transfer a manuscript that one journal cannot offer to publish to another journal, without the author having to upload the manuscript data again. To transfer your manuscript to another EMBO Press journal using this service, please click on Link Not Available

June 23, 2023

The editorial team

EMBO Reports

To the Editor,

Please find in the attachment our re-submission entitled “Loss of Myo19 increases metastasis by enhancing microenvironmental ROS gradient and chemotaxis”. We thank the editor and the editorial team for the valuable suggestions. We have added additional experiments accordingly. We found that spheroid invasion was enhanced in Myo19 knock out (KO) spheroids with sharper ROS gradient, and treatment of catalase could inhibit the invasiveness in both wild type (WT) and Myo19 KO spheroids (Fig. 3C, and line 154-155), arguing that enhanced microenvironmental ROS gradient in Myo19-depleted spheroids resulted in their increased invasiveness.

We feel that the additional data have significantly strengthened the conclusions of this manuscript and would like to re-submit it for potential peer review. We deeply appreciate your efforts in handling our manuscript.

Yours sincerely,

Congying Wu

Associate Professor

Institute of Systems Biomedicine,

Peking University Health Science Center,

Beijing, China

Dear Prof. Wu,

Thank you for the submission of your research manuscript to our journal, which was now seen by three referees, whose reports are copied below.

We concur with the referees that the proposed role of microenvironmental ROS gradient in tumor cell chemotaxis mediated by Src kinase is in principle interesting. However, the referees also raise significant concerns that need to be addressed to consider publication here.

Should you be able to address all referee concerns, we would like to invite you to submit a revised manuscript. Please revise your manuscript with the understanding that the referee concerns (as in their reports) must be fully addressed and their suggestions taken on board. Please address all referee concerns in a complete point-by-point response. Acceptance of the manuscript will depend on a positive outcome of a second round of review. It is EMBO reports policy to allow a single round of major experimental revision only and acceptance or rejection of the manuscript will therefore depend on the completeness of your responses included in the next, final version of the manuscript.

We realize that it is difficult to revise to a specific deadline. In the interest of protecting the conceptual advance provided by the work, we recommend a revision within 3 months. Please discuss the revision progress ahead of this time with me if you require more time to complete the revisions, or if you have questions or comments regarding the revision (also by video chat).

1. A data availability section providing access to data deposited in public databases is missing (where applicable).
2. Your manuscript contains statistics and error bars based on $n=2$. Please use scatter plots in these cases.

You can submit the revision either as a Scientific Report or as a Research Article. For Scientific Reports, the revised manuscript can contain up to 5 main figures and 5 Expanded View figures, and it should not exceed 27000 characters. If the revision leads to a manuscript with more than 5 main figures it will be published as a Research Article. In this case the Results and Discussion section should be separate. If a Scientific Report is submitted, these sections have to be combined. This will help to shorten the manuscript text by eliminating some redundancy that is inevitable when discussing the same experiments twice. In either case, all materials and methods should be included in the main manuscript file.

4) a .docx formatted letter INCLUDING the reviewers' reports and your detailed point-by-point responses to their comments. As part of the EMBO publication's Transparent Editorial Process, EMBO reports publishes online a Review Process File (RPF) to accompany accepted manuscripts. This File will be published in conjunction with your paper and will include the referee reports, your point-by-point response and all pertinent correspondence relating to the manuscript.

<https://www.embopress.org/page/journal/14693178/authorguide#transparentprocess>

You are able to opt out of this by letting the editorial office know (emboreports@embo.org). If you do opt out, the Review

Process File link will point to the following statement: "No Review Process File is available with this article, as the authors have chosen not to make the review process public in this case."

5) a complete author checklist, which you can download from our author guidelines

<https://www.embopress.org/page/journal/14693178/authorguide>. Please insert information in the checklist that is also reflected in the manuscript. The completed author checklist will also be part of the RPF.

6) Please note that all corresponding authors are required to supply an ORCID ID for their name upon submission of a revised manuscript (<<https://orcid.org/>>). Please find instructions on how to link your ORCID ID to your account in our manuscript tracking system in our Author guidelines

<<https://www.embopress.org/page/journal/14693178/authorguide#authorshipguidelines>>

7) Before submitting your revision, primary datasets produced in this study need to be deposited in an appropriate public database (see <https://www.embopress.org/page/journal/14693178/authorguide#datadeposition>). Please remember to provide a reviewer password if the datasets are not yet public. The accession numbers and database should be listed in a formal "Data Availability" section placed after Materials & Method (see also

<https://www.embopress.org/page/journal/14693178/authorguide#datadeposition>). Please note that the Data Availability Section is restricted to new primary data that are part of this study. * Note - All links should resolve to a page where the data can be accessed. *

Additional information on source data and instruction on how to label the files are available:

<https://www.embopress.org/page/journal/14693178/authorguide#sourcedata>

9) Our journal encourages inclusion of *data citations in the reference list* to directly cite datasets that were re-used and obtained from public databases. Data citations in the article text are distinct from normal bibliographical citations and should directly link to the database records from which the data can be accessed. In the main text, data citations are formatted as follows: "Data ref: Smith et al, 2001" or "Data ref: NCBI Sequence Read Archive PRJNA342805, 2017". In the Reference list, data citations must be labeled with "[DATASET]". A data reference must provide the database name, accession number/identifiers and a resolvable link to the landing page from which the data can be accessed at the end of the reference. Further instructions are available at <http://www.embopress.org/page/journal/14693178/authorguide#referencesformat>

10) Regarding data quantification (see Figure Legends:

<https://www.embopress.org/page/journal/14693178/authorguide#figureformat>)

12) Please also note our reference format:

I look forward to seeing a revised version of your manuscript when it is ready. Please let me know if you have questions or comments regarding the revision.

Kind regards,

Deniz Senyilmaz Tiebe

Deniz Senyilmaz Tiebe, PhD
Editor
EMBO Reports

Referee #1:

This work represents the natural continuation of a recent investigation published by authors in Nature Communication ("Mechanical instability generated by Myo19 contributes to mitochondria cristae architecture and OXPHOS" and PMID: 35562374), in which they describe the relationship between Myo19 and mitochondrial metabolism. Here, the authors want to demonstrate that these effects mediated by Myo19 control tumor invasion and metastasis, particularly by regulating the tumor microenvironmental ROS. This new study is fascinating and may extend the knowledge of the important role of ROS microenvironment in tumor growth and, especially, in metastasis.

However, I would ask the authors to perform some experiments to strengthen their data.

One concern is regarding the *in vivo* experiments reported in Figure 1. The authors injected mice with 4T1 scramble, Mic60 KD, and Myo19 KD cells. As reported in the text, they affirmed that myo19 KO led to higher invasiveness (Fig 1E). However, in the graph, it is the Mic60 KD that gives the higher rate of invasiveness. Myo19KD has a low significance of $p=0.0405$. This also occurs in Fig. 1F. Have the authors inverted the graphs? If the label is correct, authors should explain the choice to focus their investigation on myo19. Throughout the text, the authors tried to justify this only at line 118, but this is a weak connection that further experiments should validate.

Furthermore, why did the authors not investigate the contribution of Mic60 throughout the manuscript? If Mic60 is important for ROS production and mitochondrial modeling, why did the authors not measure its levels in human samples? Another thing that is not clear is that even if they found significant increases in the invasiveness, the authors did not find variations in the tumor growth. This is very strange, and the authors should justify this.

To be sure about all these *in vivo* results, I encourage authors to perform a metastatic mouse model in which the primary tumor is removed. Moreover, can the metastatic potential reduce if mice are injected with cells overexpressing myo19?

In line with this, using a 3D system is a good model to study invasiveness. However, they are "produced" by immortalized cells. Can some key experiments be conducted using metastatic cells obtained from *in vivo* experiments?

Moreover, the mitochondrial distribution, the levels of key proteins of the study (such as Src), the metabolism, the ROS abundance, and the levels of ROS scavenger factors (only to cite a few) should also be detected in samples obtained from *in vivo* experiments. This will greatly improve the quality of the investigation.

Another critical point is that the authors investigate mitochondrial dynamics and metabolism. I agree with them that their previous work deeply investigates the relationship between mitochondria and myo19. However, they should verify the activity and the dynamics of mitochondria in the new experimental conditions reported in this work (experiments reported in S3 are insufficient to support this).

Similarly, authors should further validate the importance of ROS by increasing and decreasing their levels. The authors performed experiments with antimycin only in Fig S3J and 2H. Experiments should also be conducted in other experimental conditions. As reported above, authors should perform experiments by decreasing ROS species.

Minor points:

It is kindly suggested to perform experiments detecting another important mtROS, such as superoxide anion; it is recommended to perform all experiments in at least another breast cancer cell line.

4A is performed only on MDA-MB-231 cells, while for Figure 4 B-C-D-F, there is no evidence about cell lines.

Experiments on 4F are performed on B16-F10 without a justification.

It would be better to use a more specific inhibitor for Src kinase and not PP2, which specifically inhibits Lck, Fyn, and Hck. The authors can be used Dasatinib;

4J concerning the quantification of cell number in the low chamber using the Boyden transwell assay, the sample treated with PP2 showed far too low levels of migration compared to the Src KD condition. However, it would be better to demonstrate the

inhibition of Src activity with a simple WB of the active form of Src (SrcY419).

Have authors performed invasion experiments without serum in the culture media? This is not specified in the methods. Usually, invasion should be performed without serum to exclude the effects of cellular proliferation. Furthermore, DMEM and serum have antioxidant properties.

It is of great interest to have found an important molecular mechanism regulating the ROS microenvironment. It should be useful to verify whether this mechanism can also regulate another main hallmark for cancer and metastasis, which is the regulation of cell death programs.

Referee #2:

Ren et al. report an interesting relationship between tumor ROS, SRC and cell migration in their manuscript titled "Loss of Myo19 increases metastasis by enhancing microenvironmental ROS gradient and chemotaxis". While this manuscript contains convincing experiments, I pose a number of questions and comments to the authors:

In figure 1, it would be more convincing to determine if 4t1 tumors metastasized to the lung. Outgrowth and metastasis are not necessarily the same thing. Being that "metastasis" is the title, experimental manipulation of Myo19 would need to be shown to affect metastasis.

How was the "relative H₂O₂ level" quantified in s3F, similar to 4b (S0038, Beyotime)? Have you considered mitoPy1 (https://www.tocris.com/products/mitopy1_4428)?

Is Src the only kinase sensitive to H₂O₂?

Why does Myo19 downregulation increase ROS?

Do other perturbations to actin homeostasis increase H₂O₂-induced chemotaxis? If this migration is actin independent, is this amoeboid migration?

Referee #3:

In this study by Wu and colleagues, the authors propose three major findings indicated by their experiments: (i) the identification of a ROS gradient in tumor cell spheroids which projects into a H₂O₂ gradient in the tumor microenvironment; the induction by the H₂O₂ gradient of tumor cells chemotaxis by activation of Src kinase followed by RhoA GTPase inhibition; the absence of mitochondria cristae remodeling proteins (including actin motor Myosin 19) enhances ROS gradients and promotes spheroid invasion.

The data are clearly presented, but a number of questions are raised by the interpretation, and sometime overinterpretation of the results, which should be carefully addressed.

Is the difference reported in Fig. 1B statistically significant? If yes, statistical analysis employed to show significance should be described. If not, this should be clearly mentioned in the text. The term "propensity" is inappropriate and misleading. Statistical analysis/significance is also missing for data presented in Fig. 1F, making description confusing and interpretation of the results unclear.

At page 5, authors write: "Together, we postulated that loss of cristae remodeling proteins might enhance ROS gradient in tumor spheroid periphery by increasing mitochondria ROS production." It would be a strong support to this hypothesis to show that mitochondria isolated from either control of Myo19 KD cells do indeed show differences in ROS production.

At page 6 it is stated: "Collectively, these results demonstrated the existence of a peripheral ROS gradient in 3D tumor spheroids that could project into a microenvironment H₂O₂ gradient and can be enhanced by loss of cristae integrity." Again, this overinterpretation of the results shown would be strongly supported by a demonstration of the loss of cristae integrity/decrease in modified cells by comparative high-resolution microscopy or EM.

Page 6: "We found that 3D invasion was enhanced in Myo19 KO while decreased in Nrf2 KD spheroids (Supplementary Fig. 6B), and the addition of catalase could inhibit the invasiveness in both WT and Myo19 KO spheroids (Fig. 3C)." For Figure S6, the presentation of representative images for the different experimental conditions should be shown.

End of page 6, cell-autonomous versus non-autonomous effects. The experiments shown in Fig. 3D-G are not convincing. The authors should use a simpler Matrigel-transwell invasion assay to (possibly) exclude cell-autonomous effects on 3D migration.

Page 7: "Consistently, WT+KO spheroids displayed enhanced ROS gradient and outspread faster (Fig. 3J and 3K)." An important control is missing: could this increase be reverted by abolishing the ROS gradient in this assay?

Immunochemical data in Fig. 4H and 4I: statistically significant differences from more experiments should be presented. Overall, the development of this part is weak and incomplete as it stands.

Referee #1:

This work represents the natural continuation of a recent investigation published by authors in Nature Communication (Mechanical instability generated by Myo19 contributes to mitochondria cristae architecture and OXPHOS" and PMID: 35562374), in which they describe the relationship between Myo19 and mitochondrial metabolism. Here, the authors want to demonstrate that these effects mediated by Myo19 control tumor invasion and metastasis, particularly by regulating the tumor microenvironmental ROS. This new study is fascinating and may extend the knowledge of the important role of ROS microenvironment in tumor growth and, especially, in metastasis.

We thank the reviewer for acknowledging our work. We have made extensive efforts in including experiments in our updated manuscript according to reviewer's insightful suggestions. We feel that this revised manuscript is much improved.

However, I would ask the authors to perform some experiments to strengthen their data. One concern is regarding the in vivo experiments reported in Figure 1. The authors injected mice with 4T1 scramble, Mic60 KD, and Myo19 KD cells. As reported in the text, they affirmed that myo19 KO led to higher invasiveness (Fig 1E). However, in the graph, it is the Mic60 KD that gives the higher rate of invasiveness. Myo19KD has a low significance of $p=0.0405$. This also occurs in Fig. 1F. Have the authors inverted the graphs?

We thank the reviewer for the questions. We did not invert the graphs in Fig. 1F, and Mic60 knockdown (KD) indeed led to higher invasiveness than Myo19 KD, and they both exhibited higher invasion than the control group. One reason to explain this would be that Mic60 KD may lead to much severer damage on mitochondria and influence mitochondria ROS production to a larger extent. It has been reported that Mic60 KD led to very severe cristae disruption, such as onion-shaped cristae ^[1], while Myo19 KO mitochondria also showed lower cristae frequency and displayed onion-shaped cristae only occasionally ^[2] (Fig. R1.1A). In fact, ROS gradient was more significantly enhanced in Mic60 KD rather than Myo19 KO spheroids (Fig. R1.1B and 1C). These may explain the difference in invasiveness of Mic60 KD and Myo19 KD tumors but would require future investigations. We have also included these results in our revised manuscript (Fig. EV1H).

Figure for referees not shown.

If the label is correct, authors should explain the choice to focus their investigation on myo19. Throughout the text, the authors tried to justify this only at line 118, but this is a weak connection that further experiments should validate. Furthermore, why did the authors not investigate the contribution of Mic60 throughout the manuscript? If Mic60 is important for ROS production and mitochondrial modeling, why did the authors not measure its levels in human samples?

We thank the reviewer for these questions. As the reviewer stated, this work represents the natural continuation of our recent investigation. Our lab has long been interested in cytoskeleton. Myo19 is so far the only myosin that is reported to primarily localize on mitochondria^[3], but its pathological significance is poorly understood. The comparison between Mic60 and Myo19 KD further supported that the cellular and spheroid phenotypes by Myo19 KD was associated with its role in cristae formation. However, we agreed with the reviewer that the impact of Mic60 in ROS production and tumor progression is of great interest for future studies.

During revision, we performed immunohistochemical staining of Mic60 in breast carcinoma patient samples to broaden our conclusion (Fig. R1.2A). We found that lower Mic60 expression was frequently observed in both Metastasis and Non metastasis groups (Fig. R1.2B). In addition, no significant difference in lymphatic metastasis was detected between Mic60 low expression group (L) and moderate expression group (M) (Fig. R1.2C), arguing that Mic60 expression in human breast carcinoma may not be associated with their metastatic status. We further employed the fat mammary pad injection model, and monitored lung metastases of 4T1 breast carcinoma (Fig. R1.2D). Mic60 KD increased the number of lung metastatic foci (Fig. R1.2E and 2F). Together, these results suggested that Mic60 depletion may promote the metastasis of breast carcinoma in mice but not humans.

Figure for referees not shown.

Another thing that is not clear is that even if they found significant increases in the invasiveness, the authors did not find variations in the tumor growth. This is very strange, and the authors should justify this.

We thank the reviewer for pointing this out. Previous studies indicated that tumor invasion is not necessarily accompanied by significantly enhanced tumor growth ^[4]. Invasion can occur with a small tumor cell number, while large tumors that do not show invasion can also be detected ^[5]. Here, the tumor cells may not necessarily proliferate more before invading surrounding tissues. Thus, we were able to detect the difference in tumor invasiveness without observing the difference in tumor growth. This phenomenon also suggest that the invasion observed is due to altered cell migration behavior such as chemotaxis, rather than cell growth change.

To be sure about all these in vivo results, I encourage authors to perform a metastatic mouse model in which the primary tumor is removed.

We thank the reviewer for this suggestion. During revision, we employed the fat mammary pad injection model, and monitored lung metastasis of 4T1 breast carcinoma (Fig. R1.2D). Consistent with increased invasiveness, we found increased metastatic foci in lungs of Myo19 KD and Mic60 KD tumors (Fig. R1.2E and 2F), arguing that loss of Myo19 could promote tumor metastasis. We have also included this result in our current manuscript (Fig. 1G and 1H).

Moreover, can the metastatic potential reduce if mice are injected with cells overexpressing myo19?

We thank the reviewer for the suggestion on Myo19 overexpression. Commonly, the phenotype in gene knockout cells can be rescued by gene overexpression. However, mitochondria dysfunctions such as mitochondria fragmentation ^[6], tadpole-shaped mitochondria ^[3] and perinuclear accumulation of mitochondria ^[7] are observed in Myo19 overexpressing cells. These dysfunctions were tightly linked to tumor initiation and progression. These observations suggest that Myo19 overexpression may not reduce metastatic potential.

In line with this, using a 3D system is a good model to study invasiveness. However, they are "produced" by immortalized cells. Can some key experiments be conducted using metastatic cells obtained from *in vivo* experiments?

We thank the reviewer for the suggestions. In our current manuscript, only the samples obtained from breast carcinoma patients are not immortalized (Fig. 1A-1C, Myo19 immunohistochemical staining). We agree with the reviewer that using more primary cells obtained from *in vivo* tumors would strengthen our conclusion. However, within the revision period, we were unable to obtain the protocol approval for the use of metastatic samples from patients for many of the live cell imaging and migration assays. We apologize for this and would like to investigate non-immortalized tumor cells in depth in our future work.

Moreover, the mitochondrial distribution, the levels of key proteins of the study (such as Src), the metabolism, the ROS abundance, and the levels of ROS scavenger factors (only to cite a few) should also be detected in samples obtained from *in vivo* experiments. This will greatly improve the quality of the investigation.

We thank the reviewer for the suggestions. As stated above, we have not yet been approved to perform these experiments using breast carcinoma samples in patients, but we added experiments using cells obtained from 4T1 breast carcinoma in mice. We found increased H₂O₂ level and superoxide dismutase (SOD) activity in Myo19 KD tumor lysis (Fig. R1.3A and 3B), consistent with what was found in MDA-MB-231 Myo19 KO cells. In addition, phosphorylated Src (Y419) was also upregulated in tumor tissues in response to H₂O₂ stimulation (Fig. R1.3C), arguing that H₂O₂ also activated the Src activity in tumors obtained from *in vivo* experiments.

Figure for referees not shown.

Another critical point is that the authors investigate mitochondrial dynamics and metabolism. I agree with them that their previous work deeply investigates the relationship between mitochondria and myo19. However, they should verify the activity and the dynamics of mitochondria in the new experimental conditions reported in this work (experiments reported in S3 are insufficient to support this).

We thank the reviewer for the suggestions. During revision, we examined the crista morphology using electron microscopy in our current experimental conditions as the reviewer suggested, and found that Myo19 absence decreased the crista frequency (Fig R1.1A). We have also included this result in our current manuscript (Fig. EV1H).

Similarly, authors should further validate the importance of ROS by increasing and decreasing their levels. The authors performed experiments with antimycin only in Fig S3J and 2H. Experiments should also be conducted in other experimental conditions. As reported above, authors should perform experiments by decreasing ROS species.

We thank the reviewer for the suggestions. To investigate the ROS gradient in spheroids with decreased mitochondria ROS, we treated the spheroids with MitoTEMPO to scavenge mitochondria ROS, and found that the ROS gradient was decreased (Fig. R1.4). Considering that Antimycin A treatment promoted the establishment of ROS gradient, Myo19 KO and Mic60 KD spheroids also harbored enhanced ROS gradient, these results indicated that mitochondria ROS contributed to the establishment of ROS gradient. We have included these results in the revised manuscript (Fig. 2I and Figure EV2R).

Figure for referees not shown.

Minor points:

It is kindly suggested to perform experiments detecting another important mtROS, such as;

We thank the reviewer for the suggestions on detecting another mitochondria ROS. During revision, we investigated the level of mitochondria superoxide anion in WT and Myo19 KO cells using MitoSOX probe. We found that the relative MitoSOX intensity was upregulated in Myo19 KO cells (Fig. R1.5), suggesting that Myo19 depletion upregulated mitochondria superoxide anion level. We have also included this result in our current manuscript (Figure EV2G).

Figure for referees not shown.

it is recommended to perform all experiments in at least another breast cancer cell line.

We thank the reviewer for the suggestions on repeating our experiments in another breast cancer cell line. The existence of ROS gradient in tumor spheroids composed of other tumor cell lines including HeLa, B16-F10 and MCF7, has been confirmed in our last manuscript (Fig. 2D and 2E). In addition to this, we now found that spheroids of 4T1 mouse breast carcinoma cells also displayed the ROS gradient (Fig. R1.6A and 6B). Myo19 KD could also upregulate the cellular ROS level (Fig. R1.6C) and mitochondria

ROS level (Fig. R1.6D) in 4T1 cells. Moreover, the invasion of 4T1 spheroids could also be expedited by Myo19 KD (Fig. R1.6E), while the cell-autonomous factors such as migratory ability was not affected, as measured by transwell assay (Fig. R1.6F). Last but not least, we investigated the chemotactic behavior in 4T1 and MCF7 cells, and found extracellular H₂O₂ gradient could also induce chemotaxis in these cell lines (Fig. R1.6G).

Figure for referees not shown.

4A is performed only on MDA-MB-231 cells, while for Figure 4 B-C-D-F, there is no evidence about cell lines.

We apologized for missing this information in the legends. Figure 4A-4D were performed using MDA-MB-231 cells, and Figure 4E-4F using B16-F10 cells. We have also included this information in our updated manuscript.

Experiments on 4F are performed on B16-F10 without a justification.

We thank the reviewer for pointing out this. We employed the imaging based ibidi chemotactic chamber (Fig. 4E-4F) to exclude the cell proliferation artifacts in transwell assays. Since MDA-MB-231 cells harbored higher proliferation rate, and that the proliferating cells were not analyzed due to their halted migration before division and transiently peaked velocity right after cytokinesis, we used the B16-F10 cells instead. B16-F10 cells showed high migration speed and metastatic potential^[8], which were suitable for this device and subsequent tracking and analysis. We have also added these explanations in the revised manuscript (Page 8, Line 205-209).

It would be better to use a more specific inhibitor for Src kinase and not PP2, which specifically inhibits Lck, Fyn, and Hck. The authors can use Dasatinib;

We thank the reviewer for this suggestion. During revision, we evaluated the H₂O₂-induced chemotaxis in cells treated with Dasatinib using transwell assay. Consistent with Src knock down and PP2 treatment, Dasatinib also inhibited cell migration towards the lower chamber upon H₂O₂ gradient, indicative of inhibited H₂O₂-induced chemotaxis (Fig. R1.7) We have also included this result in our updated manuscript (Fig. 4L).

Figure for referees not shown.

4J concerning the quantification of cell number in the low chamber using the Boyden transwell assay, the sample treated with PP2 showed far too low levels of migration compared to the Src KD condition. However, it would be better to demonstrate the inhibition of Src activity with a simple WB of the active form of Src (SrcY419).

We thank the reviewer for the suggestions. We took the advice and also included Dasatinib, the more specific Src inhibitor that the reviewer suggested. We assayed the level of phosphorylated Src Y419 using Western blot, and found downregulated active Src in cells treated with PP2 or Dasatinib (Fig. R1.8), suggesting that PP2 and Dasatinib treatments in the transwell assay indeed inhibited Src activity. We have also included these results in the revised manuscript (Figure EV5F).

Figure for referees not shown.

Have authors performed invasion experiments without serum in the culture media? This is not specified in the methods. Usually, invasion should be performed without serum to exclude the effects of cellular proliferation. Furthermore, DMEM and serum have antioxidant properties.

We thank reviewer for this suggestion. We assayed the 3D invasion of WT and Myo19 KO spheroids in serum-free culture media, and still found higher invasion rate in spheroids

depleted of Myo19 (Fig. R1.9), suggesting that Myo19 absence promoted spheroid invasiveness.

Figure for referees not shown.

It is of great interest to have found an important molecular mechanism regulating the ROS microenvironment. It should be useful to verify whether this mechanism can also regulate another main hallmark for cancer and metastasis, which is the regulation of cell death programs.

We thank the reviewer for acknowledging our work. ROS, which are considered as double-edge swords, are involved in tumor initiation, proliferation, invasion and angiogenesis^[9]. But the excessive ROS would also trigger senescence and multiple cell death pathways^[10-12].

The inner core of tumor spheroids and solid tumors are characterized of various cell death pathways, namely apoptosis, necrosis and ferroptosis^[13-16]. Treatment of antioxidants reduced the cell death in the inner spheroids^[14,16], suggesting that cell death in spheroids is associated with their redox environment. It is intriguing to investigate how microenvironmental ROS regulated the cell death pathways in tumor spheroids, and whether mitochondria ROS was involved in this regulation. We have added a section of discussion on this topic (Page 9-10, Line 244-250).

Referee #2:

Ren et al. report an interesting relationship between tumor ROS, SRC and cell migration in their manuscript titled "Loss of Myo19 increases metastasis by enhancing microenvironmental ROS gradient and chemotaxis". While this manuscript contains convincing experiments, I pose a number of questions and comments to the authors:

We thank the reviewer for acknowledging our work. We have included supplemental experiments to broaden our conclusion according to reviewer's insightful comments and suggestions. We feel that this revised manuscript is much improved.

In figure 1, it would be more convincing to determine if 4t1 tumors metastasized to the lung. Outgrowth and metastasis are not necessarily the same thing. Being that "metastasis" is the title, experimental manipulation of Myo19 would need to be shown to affect metastasis.

We thank the reviewer for this suggestion. During revision, we employed the fat mammary pad injection model, and monitored lung metastasis of 4T1 breast carcinoma (Fig. R2.1A). Consistent with increased invasiveness, we found increased metastatic foci in lungs of Myo19 knock down (Myo19 KD) and Mic60 knock down (Mic60 KD) tumors (Fig. R2.1B and 1C), arguing that loss of Myo19 and Mic60 could promote tumor metastasis. We have also included these results in our current manuscript (Fig. 1G and 1H).

Figure for referees not shown.

How was the "relative H₂O₂ level" quantified in s3F, similar to 4b (S0038, Beyotime)?

We thank the reviewer for pointing this out. "H₂O₂ level detection kit" (S0038, Beyotime) can be used to detect H₂O₂ level in cell lysis and culture media. In this assay, H₂O₂ in the samples could oxidize Fe²⁺ to Fe³⁺ in the detection reagent, which further reacted with the xylenol orange dye to yield product with maximum absorbance at 560 nm. In Fig. S3F, we inspected the H₂O₂ level in WT and Myo19 knock out (Myo19 KO) cell lysis, and in Fig. 4B we measured H₂O₂ level of culture media from upper and lower chambers in transwell assay. We calculated the "relative H₂O₂ level" to normalize the experimental values. To do this, we first calculated the average H₂O₂ level of control group (WT group in Fig. S3F and the upper chamber group in Fig. 4B), and then divided the experimental values in each group with this average value. We have also included the analysis in our updated methods session (Page 17, Line 435-437).

Have you considered mitoPy1 (https://www.tocris.com/products/mitopy1_4428)?

We thank the reviewer for this suggestion. During revision, we employed the mitoPY1 to measure H₂O₂ level in WT and Myo19 KO cells. We found increased mitoPY1 intensity in Myo19 KO cells (Fig. R2.2), indicating upregulated H₂O₂ level. We have also included this result in our revised manuscript (Figure EV2J).

Figure for referees not shown.

Is Src the only kinase sensitive to H₂O₂?

We thank the reviewer for this question. Some other protein tyrosine kinases have also been reported sensitive to redox stimulation, such as the Src family proteins Fyn and Lyn. We focused on Src as a candidate because it is not only sensitive to H₂O₂ stimulation through direct cysteine oxidation ^[17], but also well-known to regulate cell adhesion and migration ^[18]. Notably, the oxidatively activated Src has been reported to promote cell migration ^[19,20] and the role of Src in chemotaxis has also been demonstrated ^[21]. These prompted us to study Src. We would like to further screen for H₂O₂ chemotactic “sensors” in our future work.

Why does Myo19 downregulation increase ROS?

We thank the reviewer for pointing this out. Mitochondria are one of the major sources for cellular ROS production, and mitochondria ROS are produced during OXPHOS ^[22], which takes place at cristae. Absence of cristae sculpturing proteins such as Myo19 and Mic60 disrupts cristae, which enhances electron leakage and promotes ROS production ^[1,23]. We apologize for missing this rationale in the previous manuscript, and we have added this in the revised manuscript (Page 4, Line 98-99). During revision, we have also examined mitochondria fine structure with electron microscopy (Fig. R2.3) and verified that loss of Myo19 or Mic60 disrupted cristae structure, which may then increase mitochondria ROS. We have also included the electron microscopy data in our revised manuscript (Figure EV1H).

Figure for referees not shown.

Do other perturbations to actin homeostasis increase H₂O₂-induced chemotaxis? If this migration is actin independent, is this amoeboid migration?

We thank the reviewer for the suggestions. The role of actin in directed migration has been extensively studied ^[24,25]. We employed transwell assay and investigated the chemotaxis of cells treated with Latrunculin B (LatB), which inhibits actin polymerization. Low concentration of LatB did not completely abolish cell migration towards the lower chamber but partially inhibited the chemotactic efficiency (Fig. R2.4A). However, global disruption of the actin cytoskeleton by LatB may affect chemotaxis through inhibiting intrinsic migration machineries other than altering the H₂O₂ chemotaxis.

Actin plays an important role in both mesenchymal and amoeboid migration. To evaluate whether H₂O₂-induced chemotaxis was mesenchymal or amoeboid migration, we assayed the 2D migration of cells within the H₂O₂ gradient. We found the migrating cells adopted a well-spread morphology and displayed strong adhesion with the substrate (Fig. R2.4B), which is one of the characteristics of mesenchymal migration ^[26]. This result indicated that in our experimental condition, H₂O₂-induced chemotaxis is likely to be mesenchymal migration.

Figure for referees not shown.

Referee #3:

In this study by Wu and colleagues, the authors propose three major findings indicated by their experiments: (i) the identification of a ROS gradient in tumor cell spheroids which projects into a H₂O₂ gradient in the tumor microenvironment; the induction by the H₂O₂ gradient of tumor cells chemotaxis by activation of Src kinase followed by RhoA GTPase inhibition; the absence of mitochondria cristae remodeling proteins (including actin motor Myosin 19) enhances ROS gradients and promotes spheroid invasion. The data are clearly presented, but a number of questions are raised by the interpretation, and sometime overinterpretation of the results, which should be carefully addressed.

We thank the reviewer for acknowledging our work. We have included supplemental experiments and modified the text to avoid misleading statements in our updated manuscript accordingly. We have also modified the manuscript to exclude overinterpretation of the results. We feel that this revised manuscript is much improved.

Is the difference reported in Fig. 1B statistically significant? If yes, statistical analysis employed to show significance should be described. If not, this should be clearly mentioned in the text. The term "propensity" is inappropriate and misleading.

We apologize for the inappropriate and misleading descriptions. The statistical analysis of Fig. 1B was conducted and shown in Fig. 1C. To avoid misunderstandings, we have deleted the word "propensity" and more specifically interpreted this result in the revised manuscript (Page 3, Line 71). Fig. 1B displayed the probability distribution of Myo19 expression in the two groups, and revealed a left shift of the peak value in the Metastasis group, which indicated a higher metastatic frequency in the lower Myo19 expression samples. Next, we sought to divide the samples into Myo19 Low, Moderate and High groups, and found higher frequency of metastatic incidents in the Low group compared to the Moderate group (Fig. 1C). Together, these results suggested that lower-expressed Myo19 in tumors was associated with higher metastasis.

Statistical analysis/significance is also missing for data presented in Fig. 1F, making description confusing and interpretation of the results unclear.

We apologize for the confusing descriptions. Loss of Myo19 promoted the local invasion of mouse breast carcinoma into normal skeletal muscles, and loss of Mic60 promoted the invasion into skeletal muscle and skin tissues. To more clearly displayed these results and avoid misinterpretations, we replaced the graph in Fig. 1F with a table to elucidate the sites where the tumor cells were invading, and added the statistical significance below the number (Fig. 1F).

Percentage (%)	scramble	Mic60 KD	Myo19 KD
Non invasion	28	0	16

Skeletal Muscle	0	57 (****)	66 (****)
Epidermis	14	57 (****)	16 (ns)
Dermis	71	71 (****)	33 (ns)

At page 5, authors write: "Together, we postulated that loss of cristae remodeling proteins might enhance ROS gradient in tumor spheroid periphery by increasing mitochondria ROS production." It would be a strong support to this hypothesis to show that mitochondria isolated from either control of Myo19 KD cells do indeed show differences in ROS production.

We thank the reviewer for the suggestions. During revision, we isolated mitochondria in WT and Myo19 KO cells (Fig. R3.1A), and assayed the H₂O₂ level in the lysis of isolated mitochondria as the reviewer suggested. We found that Myo19 KO mitochondria displayed higher H₂O₂ level (Fig. R3.1B). In addition, we investigated the level of mitochondria superoxide anion in WT and Myo19 KO cells using MitoSOX. We found that the MitoSOX intensity was upregulated in Myo19 KO cells (Fig. R3.1C). Together, these results strengthened the notion that Myo19 KO induced mitochondria ROS production. We have also included these results in our revised manuscript (Figure EV2E-2G).

Figure for referees not shown.

At page 6 it is stated: "Collectively, these results demonstrated the existence of a peripheral ROS gradient in 3D tumor spheroids that could project into a microenvironment H₂O₂ gradient and can be enhanced by loss of cristae integrity." Again, this

overinterpretation of the results shown would be strongly supported by a demonstration of the loss of cristae integrity/decrease in modified cells by comparative high-resolution microscopy or EM.

We apologize for this overinterpretation and thank the reviewer for the suggestions. It has been reported by us and others that absence of Myo19 or Mic60 could disrupt cristae integrity, as indicated by electron microscopy ^[1,2,27] and Hessian-SIM super resolution microscopy ^[28]. To further strengthen this conclusion, we re-inspected the crista morphology in WT, Myo19 KO and Mic60 KD cells using electron microscopy in our current experimental conditions as the reviewer suggested. We found that absence of Myo19 or Mic60 could affect cristae morphology (Fig R3.2). We have also included this result in our revised manuscript (Figure EV1H).

Figure for referees not shown.

Page 6: "We found that 3D invasion was enhanced in Myo19 KO while decreased in Nrf2 KD spheroids (Supplementary Fig. 6B), and the addition of catalase could inhibit the invasiveness in both WT and Myo19 KO spheroids (Fig. 3C)." For Figure S6, the presentation of representative images for the different experimental conditions should be shown.

We apologize for missing these representative images and thank the reviewer for the suggestions. We have added the representative images of the 3D invasion of WT/Myo19

KO/Nrf KD spheroids (Supplementary Fig. 6B in the last manuscript and Figure EV7C in the revised manuscript) and WT/Myo19 KO spheroids under catalase treatment (Fig. 3C) in the revised manuscript (Fig. R3.3, Figure EV4B).

Figure for referees not shown.

End of page 6, cell-autonomous versus non-autonomous effects. The experiments shown in Fig. 3D-G are not convincing. The authors should use a simpler Matrigel-transwell invasion assay to (possibly) exclude cell-autonomous effects on 3D migration.

We thank the reviewer for the suggestions. During revision, we assayed WT and Myo19 KO cell invasion using Matrigel-coated transwells and found that the migratory cells to the lower chamber remained steady upon Myo19 depletion (Fig. R3.4), which argued against cell-autonomous effects on 3D migration.

Figure for referees not shown.

Page 7: "Consistently, WT+KO spheroids displayed enhanced ROS gradient and outspread faster (Fig. 3J and 3K)." An important control is missing: could this increase be reverted by abolishing the ROS gradient in this assay?

We apologized for the ambiguous descriptions. In this assay, we did perform the control group with spheroids of abolished ROS gradient, which was the KO+WT group (with Myo19 KO preformed core and WT layered around the periphery). Reconstructed spheroids of KO+WT group harbored decreased ROS gradient since Myo19 KO cells displayed higher ROS level and WT cells displayed lower ROS. Indeed, compared to WT+WT spheroids with normal ROS gradient and invasion rate, KO+WT spheroids displayed decreased ROS gradient and slowed spreading (Fig. 2J and 2K). We have rewritten this part with more specific descriptions in the revised manuscript (Page 7, Line 182-185).

Immunochemical data in Fig. 4H and 4I: statistically significant differences from more experiments should be presented. Overall, the development of this part is weak and incomplete as it stands.

We apologize for missing this information and thank the reviewer for the suggestions. We have included the Western blot in Fig. 4H and 4I with three independent experiments as the reviewer suggested, and found that H₂O₂ stimulation upregulated phosphorylated Y419 Src level (Fig. R3.5A) and downregulated active RhoA level (Fig. R3.5B) using Student's *t*-test. These results strengthened our findings that H₂O₂ could activate Src and inhibit RhoA. Considering that Src inhibition and RhoA activation could both inhibit H₂O₂-induced chemotaxis (Fig. 4J), these results suggested that H₂O₂ could induce chemotaxis through Src-RhoA signaling. We have also included these results in our revised manuscript (Fig. 4H-4K).

Figure for referees not shown.

B. Left: immunoblotting of MDA-MB-231 cells treated without or with 200 μM H_2O_2 for 5 hours. GAPDH was used as a loading control. Right: quantification of relative active RhoA level. Data are shown as mean \pm SD. $N_{\text{ctrl}}=3$. $N_{\text{H}_2\text{O}_2}=3$. *, $p=0.0326$.

Reference:

- [1] John GB, Shang Y, Li L, Renken C, Mannella CA, Selker JM, Rangell L, Bennett MJ, and Zha J. The mitochondrial inner membrane protein mitofilin controls cristae morphology. *Mol Biol Cell*. 2005, 16 (3): 1543-54
- [2] Shi P, Ren X, Meng J, Kang C, Wu Y, Rong Y, Zhao S, Jiang Z, Liang L, He W, Yin Y, Li X, Liu Y, Huang X, Sun Y, Li B, and Wu C. Mechanical instability generated by Myosin 19 contributes to mitochondria cristae architecture and OXPHOS. *Nat Commun*. 2022, 13 (1): 2673
- [3] Quintero OA, DiVito MM, Adikes RC, Kortan MB, Case LB, Lier AJ, Panaretos NS, Slater SQ, Rengarajan M, Feliu M, and Cheney RE. Human Myo19 is a novel myosin that associates with mitochondria. *Curr Biol*. 2009, 19 (23): 2008-13
- [4] Sonoshita M, Itatani Y, Kakizaki F, Sakimura K, Terashima T, Katsuyama Y, Sakai Y, and Taketo MM. Promotion of colorectal cancer invasion and metastasis through activation of NOTCH-DAB1-ABL-RHOGEF protein TRIO. *Cancer Discov*. 2015, 5 (2): 198-211
- [5] Luzzi KJ, MacDonald IC, Schmidt EE, Kerkvliet N, Morris VL, Chambers AF, and Groom AC. Multistep nature of metastatic inefficiency: dormancy of solitary cells after successful extravasation and limited survival of early micrometastases. *Am J Pathol*. 1998, 153 (3): 865-73
- [6] Coscia SM, Thompson CP, Tang Q, Baltrusaitis EE, Rhodenhiser JA, Quintero-Carmona OA, Ostap EM, Lakadamyali M, and Holzbaur ELF. Myo19 tethers mitochondria to endoplasmic reticulum-associated actin to promote mitochondrial fission. *J Cell Sci*. 2023, 136 (5): jcs260612
- [7] Oeding SJ, Majstrowicz K, Hu XP, Schwarz V, Freitag A, Honnert U, Nikolaus P, and Bähler M. Identification of Miro1 and Miro2 as mitochondrial receptors for myosin XIX. *J Cell Sci*. 2018, 131 (17): jcs219469
- [8] Nakamura K, Yoshikawa N, Yamaguchi Y, Kagota S, Shinozuka K, and Kunitomo M. Characterization of mouse melanoma cell lines by their mortal malignancy using an experimental metastatic model. *Life Sci*. 2002, 70 (7): 791-8
- [9] Kirtonia A, Sethi G, and Garg M. The multifaceted role of reactive oxygen species in tumorigenesis. *Cell Mol Life Sci*. 2020, 77 (22): 4459-4483
- [10] Circu ML and Aw TY. Reactive oxygen species, cellular redox systems, and apoptosis. *Free Radic Biol Med*. 2010, 48 (6): 749-62
- [11] Stockwell BR, Friedmann Angeli JP, Bayir H, Bush AI, Conrad M, Dixon SJ, Fulda S, Gascón S, Hatzios SK, Kagan VE, Noel K, Jiang X, Linkermann A, Murphy ME, Overholtzer M, Oyagi A, Pagnussat GC, Park J, Ran Q, Rosenfeld CS, Salnikow K, Tang D, Torti FM, Torti SV, Toyokuni S, Woerpel KA, and Zhang DD. Ferroptosis: A Regulated Cell Death Nexus Linking Metabolism, Redox Biology, and Disease. *Cell*. 2017, 171 (2): 273-285

- [12] Kim YS, Morgan MJ, Choksi S, and Liu ZG. TNF-induced activation of the Nox1 NADPH oxidase and its role in the induction of necrotic cell death. *Mol Cell*. 2007, 26 (5): 675-87
- [13] Bell HS, Whittle IR, Walker M, Leaver HA, and Wharton SB. The development of necrosis and apoptosis in glioma: experimental findings using spheroid culture systems. *Neuropathol Appl Neurobiol*. 2001, 27 (4): 291-304
- [14] Schafer ZT, Grassian AR, Song L, Jiang Z, Gerhart-Hines Z, Irie HY, Gao S, Puigserver P, and Brugge JS. Antioxidant and oncogene rescue of metabolic defects caused by loss of matrix attachment. *Nature*. 2009, 461 (7260): 109-13
- [15] Demuyneck R, Efimova I, Lin A, Declercq H, and Krysko DV. A 3D Cell Death Assay to Quantitatively Determine Ferroptosis in Spheroids. *Cells*. 2020, 9 (3): 703
- [16] Takahashi N, Cho P, Selfors LM, Kuiken HJ, Kaul R, Fujiwara T, Harris IS, Zhang T, Gygi SP, and Brugge JS. 3D Culture Models with CRISPR Screens Reveal Hyperactive NRF2 as a Prerequisite for Spheroid Formation via Regulation of Proliferation and Ferroptosis. *Mol Cell*. 2020, 80 (5): 828-844.e6
- [17] Heppner DE, Dustin CM, Liao C, Hristova M, Veith C, Little AC, Ahlers BA, White SL, Deng B, Lam YW, Li J, and van der Vliet A. Direct cysteine sulfenylation drives activation of the Src kinase. *Nat Commun*. 2018, 9 (1): 4522
- [18] Huveneers S and Danen EH. Adhesion signaling - crosstalk between integrins, Src and Rho. *J Cell Sci*. 2009, 122 (Pt 8): 1059-69
- [19] Basuroy S, Dunagan M, Sheth P, Seth A, and Rao RK. Hydrogen peroxide activates focal adhesion kinase and c-Src by a phosphatidylinositol 3 kinase-dependent mechanism and promotes cell migration in Caco-2 cell monolayers. *Am J Physiol Gastrointest Liver Physiol*. 2010, 299 (1): G186-95
- [20] Giannoni E, Buricchi F, Raugei G, Ramponi G, and Chiarugi P. Intracellular reactive oxygen species activate Src tyrosine kinase during cell adhesion and anchorage-dependent cell growth. *Mol Cell Biol*. 2005, 25 (15): 6391-403
- [21] Ngo HT, Azab AK, Farag M, Jia X, Melhem MM, Runnels J, Roccaro AM, Azab F, Sacco A, Leleu X, Anderson KC, and Ghobrial IM. Src tyrosine kinase regulates adhesion and chemotaxis in Waldenstrom macroglobulinemia. *Clin Cancer Res*. 2009, 15 (19): 6035-41
- [22] Murphy MP. How mitochondria produce reactive oxygen species. *Biochem J*. 2009, 417 (1): 1-13
- [23] Majstrowicz K, Honnert U, Nikolaus P, Schwarz V, Oeding SJ, Hemkemeyer SA, and Bähler M. Coordination of mitochondrial and cellular dynamics by the actin-based motor Myo19. *J Cell Sci*. 2021, 134 (10): jcs255844
- [24] Weiner OD, Servant G, Welch MD, Mitchison TJ, Sedat JW, and Bourne HR. Spatial control of actin polymerization during neutrophil chemotaxis. *Nat Cell Biol*. 1999, 1 (2): 75-81
- [25] Bear JE and Haugh JM. Directed migration of mesenchymal cells: where signaling and the cytoskeleton meet. *Curr Opin Cell Biol*. 2014, 30 74-82
- [26] SenGupta S, Parent CA, and Bear JE. The principles of directed cell migration. *Nat Rev Mol Cell Biol*. 2021, 22 (8): 529-547
- [27] Stephan T, Brüser C, Deckers M, Steyer AM, Balzarotti F, Barbot M, Behr TS, Heim G,

- Hübner W, Ilgen P, Lange F, Pacheu-Grau D, Pape JK, Stoldt S, Huser T, Hell SW, Möbius W, Rehling P, Riedel D, and Jakobs S. MICOS assembly controls mitochondrial inner membrane remodeling and crista junction redistribution to mediate cristae formation. *Embo j.* 2020, 39 (14): e104105
- [28] Hu C, Shu L, Huang X, Yu J, Li L, Gong L, Yang M, Wu Z, Gao Z, Zhao Y, Chen L, and Song Z. OPA1 and MICOS Regulate mitochondrial crista dynamics and formation. *Cell Death Dis.* 2020, 11 (10): 940

Dear Prof. Wu,

Thank you for submitting your revised manuscript. It has now been seen by all of the original referees.

As you can see, the referees find that the study is significantly improved during revision and recommend publication. However, I need you to address the points below before I can accept the manuscript.

- Please address the remaining minor concern of referee #3.
- We believe your manuscript is better suited for our Scientific Report format in terms of text length and figure count (<https://www.embopress.org/page/journal/14693178/authorguide#researcharticleguide>). Therefore, I would like to encourage you to merge the Results and Discussion sections into one.
- Please provide 3-5 keywords for your study. These will be visible in the html version of the paper and on PubMed and will help increase the discoverability of your work.
- Data Availability section needs to be placed before the Acknowledgements section.
- Conflict of Interests section needs to be renamed as "Disclosure Statement and Competing Interests"
- Author contributions section needs to be removed from the manuscript text.
- As per our format requirements, in the reference list, citations should be listed in alphabetical order and then chronologically, with the authors' surnames and initials inverted; where there are more than 10 authors on a paper, 10 will be listed, followed by 'et al.'. Please see <https://www.embopress.org/page/journal/14693178/authorguide#referencesformat>
- We note that the ORCID iDs of Dr. Shi is currently missing. EMBO Press policy asks for all corresponding authors to link to their ORCID iDs. Please see "Authorship Guidelines" in the Guide to Authors here: <https://www.embopress.org/page/journal/14693178/authorguide#authorshipguidelines>

In order to link your ORCID iD to your account in our manuscript tracking system, please do the following:

1. Click the 'Modify Profile' link at the bottom of your homepage in our system.
2. On the next page you will see a box halfway down the page titled ORCID*. Below this box is red text reading 'To Register/Link to ORCID, click here'. Please follow that link: you will be taken to ORCID where you can log in to your account (or create an account if you don't have one)
3. You will then be asked to authorise Wiley to access your ORCID information. Once you have approved the linking, you will be brought back to our manuscript system.

We regret that we cannot do this linking on your behalf for security reasons.

- We note the following regarding the movie file: its name and callout need to be changed to Movie EV1; the legend should be provided as a readme.txt file and it should be zipped together with the movie and uploaded as a zipped folder titled Movie EV1.
- The source data of the main figure should be uploaded as one zip file per figure. Source data for EV figures can be grouped into one folder.
- Supplementary Figure legends on p 17 should be changed to Expanded View figure legends.
- Our production/data editors have asked you to clarify several points in the figure legends:
 - Please indicate the statistical test used for data analysis in the legends of figures 1c, e, h; 2f-i, k; 3c, e, g, j-m; 4b, d, g, i, k, l; EV1d, f, g; EV2b-c, f-j, m; EV4c-e, g; EV5c-d.
- Regarding the data from human samples, please include a statement confirming that informed consent was obtained from all subjects and that the experiments conformed to the principles set out in the WMA Declaration of Helsinki and the Department of Health and Human Services Belmont Report in the Materials and Methods section.
- Papers published in EMBO Reports include a 'synopsis' and 'bullet points' to further enhance discoverability. Both are displayed on the html version of the paper and are freely accessible to all readers. The synopsis includes a short standfirst summarizing the study in 1 or 2 sentences (max 35 words) that summarize the paper and are provided by the authors and streamlined by the handling editor. I would therefore ask you to include your synopsis blurb and 3-5 bullet points listing the key experimental findings.
- In addition, please provide an image for the synopsis. This image should provide a rapid overview of the question addressed in the study but still needs to be kept fairly modest since the image size cannot exceed 550 (width) x 300-600 (height) pixels.

Thank you again for giving us to consider your manuscript for EMBO Reports, I look forward to your minor revision.

Kind regards,

Deniz Senyilmaz Tiebe

--

Deniz Senyilmaz Tiebe, PhD
Editor

EMBO Reports

Referee #1:

I thoroughly reviewed the revised manuscript and found it to be accurate. The authors diligently incorporated my suggestions and conducted a compelling series of new in vitro and in vivo experiments, significantly enhancing the depth of the investigation. These experiments have provided robust evidence highlighting the pivotal role of Myo19 in controlling tumor growth. Moreover, they shed light on how tumor microenvironmental ROS and the disorganization of mitochondria cristae significantly influence tumor invasion and metastasis.

With these experiments, you've presented compelling evidence regarding the critical function of Myo19 and elucidated the profound impact of ROS and mitochondria cristae disorganization on tumor behavior. Given these substantial findings, I believe the manuscript merits publication in the esteemed research journal EMBO Reports.

However, I have one final request pertaining to the discussion section. While the authors adeptly focused on ROS, tumor invasion, and metastasis, I propose the inclusion of a succinct paragraph discussing the pertinent role of Myo19 in regulating the ROS-mediated effects uncovered in your research. I believe this addition will allow readers to grasp the key takeaways more effectively, particularly emphasizing the significance of your investigation in advancing our understanding of cancer and its potential therapeutic implications.

Referee #2:

Ren et al. have generated a commendable revision of excellent science. This manuscript merits publication and I commend the quality and clarity of the point-by-point rebuttal.

Congratulations,
Kevin Tharp

Referee #3:

The authors have satisfactorily answered to the points raised by this reviewer. The article is suitable for publication in E.R.

Dec. 10, 23

Deniz Senyilmaz Tiebe, PhD

Scientific Editor

EMBO Reports

Dear Dr. Tiebe,

We really appreciate your efforts in processing our manuscript. Please find in the attachment our resubmission of “Loss of Myo19 increases metastasis by enhancing microenvironmental ROS gradient and chemotaxis”. We thank you and the reviewers for critically evaluating our work. We are glad that all three reviewers found our work much improved. Here we have updated this manuscript according to editorial suggestions and the reviewers’ comments.

- We have included a succinct paragraph discussing the role of Myo19 in regulating the ROS-mediated effects as the reviewer suggested.
- We have included the key words, synopsis and bullet points in the revised manuscript to summarize our work. We also prepared an image for the synopsis.
- We have revised the manuscript according to the “Scientific Report” format requirements.

We have highlighted these changes in the updated manuscript. Please contact us if there are any remaining issues that need to be resolved.

Yours sincerely,

Congying Wu

Associate Professor

Institute of Systems Biomedicine,

Peking University Health Science Center,

Beijing, China

Dear Prof. Wu,

Thank you for submitting your revised manuscript. I have now looked at everything and all is fine. Therefore, I am very pleased to accept your manuscript for publication in EMBO Reports.

Congratulations on a nice work!

Kind regards,

Deniz Senyilmaz Tiebe

--

Deniz Senyilmaz Tiebe, PhD

Editor

EMBO Reports

--
